



# The Cloud Feedback Model Intercomparison Project (CFMIP) contribution to CMIP6.

Mark J. Webb[1], Timothy Andrews[1], Alejandro Bodas-Salcedo[1], Sandrine Bony[2], Christopher S. Bretherton[3], Robin Chadwick[1], Hélène Chepfer[2], Hervé Douville[4], Peter Good[1], Jennifer E. Kay[5], Stephen A. Klein[6], Roger Marchand[3], Brian Medeiros[7], A. Pier Siebesma[8], Christopher B. Skinner[9], Bjorn Stevens[10], George Tselioudis[11], Yoko Tsushima[1], Masahiro Watanabe[12].

[1]Met Office Hadley Centre, Exeter, United Kingdom.
[2]LMD/IPSL, CNRS, Université Pierre and Marie Curie, Paris, France.
[3]University of Washington, Seattle, USA.
[4]Centre National de Recherches Météorologiques, Toulouse, France.
[5]University of Colorado at Boulder, Boulder, USA.
[6]Lawrence Livermore National Laboratory, Livermore, USA.
[7]National Center for Atmospheric Research, Boulder, USA.
[8]Royal Netherlands Meteorological Institute, De Bilt, The Netherlands.
[9]University of Michigan, Ann Arbor, USA.
[10]Max Planck Institute for Meteorology, Hamburg, Germany.
[11]NASA Goddard Institute for Space Studies, New York, USA.
[12]Atmosphere and Ocean Research Institute, Tokyo, Japan.

Correspondence to: Mark Webb (mark.webb@metoffice.gov.uk)

Submitted to Geoscientific Model Development (GMD) 30th March, 2016.

## Abstract

The primary objective of CFMIP is to inform future assessments of cloud feedbacks through improved understanding of cloud-climate feedback mechanisms and better evaluation of cloud processes and cloud feedbacks in climate models. However, the CFMIP approach is also increasingly being used to understand other aspects of climate change, such as nonlinear change and regional changes in atmospheric circulation and precipitation. CFMIP is supporting ongoing model inter-comparison activities by coordinating a hierarchy of targeted experiments for CMIP6, along with a set of cloud related output diagnostics. CFMIP contributes primarily to addressing the CMIP6 questions "How does the Earth System respond to forcing?" and "What are the origins and consequences of systematic model biases?" and supports the activities of the WCRP Grand Challenge on Clouds, Circulation and Climate Sensitivity.

A compact set of Tier 1 experiments is proposed for CMIP6 to address the question: "1) What are the physical mechanisms underlying the range of cloud feedbacks and cloud adjustments predicted by climate models, and which models have the most credible cloud feedbacks?" Additional Tier 2 experiments are proposed to address the following questions: 2) Are cloud feedbacks consistent for climate cooling and warming, and if not, why? 3) How do cloud-radiative effects impact the structure, the strength and the variability of the general atmospheric circulation in present and future climates? 4) How do responses in the climate system due to changes in solar forcing differ from changes due to $CO_2$, and is the response sensitive to the sign of the forcing? 5) To what extent is regional climate change per $CO_2$ doubling state-dependent (nonlinear), and why? 6) Are climate feedbacks during the $20^{th}$ century different to those acting on long term climate change and climate sensitivity? 7) How do regional climate responses (e.g. in precipitation) and their uncertainties in coupled models arise from the combination of different aspects of $CO_2$ forcing and sea surface warming?

CFMIP also proposes a number of additional model outputs in the CMIP DECK, CMIP6 Historical and CMIP6 CFMIP experiments, including COSP simulator outputs and process diagnostics to address the following questions: 1) How well do clouds and other relevant variables simulated by models agree with observations? 2) What physical processes and mechanisms are important for a credible simulation of clouds, cloud feedbacks and cloud adjustments in climate models? 3) Which models have the most credible representations of processes relevant to the simulation of clouds? 4) How do clouds and their changes interact with other elements of the climate system?



## 1 Introduction

Inter-model differences in cloud feedbacks continue to be the largest source of uncertainty in predictions of equilibrium climate sensitivity (Boucher et al., 2013). Although the ranges of cloud feedbacks and climate sensitivity from comprehensive climate models have not reduced in recent years, considerable progress has been made in understanding (a) which types of clouds contribute most to this spread (e.g. Bony and Dufresne 2005; Webb et al., 2006; Zelinka et al., 2013), (b) the role of cloud adjustments in climate sensitivity (e.g. Gregory and Webb, 2008; Andrews and Forster, 2008; Kamae and Watanabe, 2012; Vial et al., 2013; Zelinka et al., 2013), (c) the processes and mechanisms which are (and are not) implicated in cloud feedbacks (e.g. Rieck et al., 2012; Brient and Bony 2012; Webb and Lock 2013; Brient and Bony 2013; Sherwood et al., 2014; Ringer et al., 2014; Medeiros et al., 2015; Bretherton et al., 2015; Zhao, 2015; Webb et al., 2015b), (d) the inconstancy of cloud feedbacks and effective climate sensitivity (e.g. Senior and Mitchell, 2000; Williams et al., 2008; Andrews et al., 2012; Geoffroy et al., 2013; Armour et al., 2013; Andrews et al., 2015), and (e) the extent to which models with stronger or weaker cloud feedbacks or climate sensitivities agree with observations (e.g. Fasullo and Trenberth, 2012; Su et al., 2014; Qu et al., 2014; Sherwood et al., 2014; Brient et al., 2015; Tsushima et al., 2015; Myers and Norris, 2016). Additionally, our ability to evaluate model clouds using satellite data has benefited from the increasing use of satellite simulators. This approach first introduced by Yu et al, 1996 for use with data from the International Satellite Cloud Climatology Project (ISCCP) attempts to reproduce what a satellite would observe given the model state. Such approaches enable more quantitative comparisons to the satellite record (e.g. Yu et al., 1996; Klein and Jakob, 1999; Webb et al.; 2001; Marchand and Ackerman, 2010; Bodas-Salcedo et al., 2011; Nam et al., 2012a; Cesana and Chepfer, 2013; Klein et al., 2013; Chepfer et al., 2014). Much of our improved understanding in these areas would have been impossible without the continuing investment of the scientific community in successive phases of the Coupled Model Intercomparison Project (CMIP), and its co-evolution in more recent years with the Cloud Feedback Model Intercomparison Project (CFMIP).

CFMIP started in 2003 and its first phase (CFMIP-1) organised an intercomparison based on perpetual July SST forced Cess style +2K experiments and $2xCO_2$ equilibrium mixed-layer model experiments containing ISCCP simulator in parallel with CMIP3 (McAvaney and Le Treut, 2003). CFMIP-1 had a substantial impact on the evaluation of clouds in models and in the identification of low level cloud feedbacks as the primary cause of inter-model spread in cloud feedback, which featured prominently in the fourth and fifth IPCC assessments (Randall et al., 2007; Boucher et al., 2013).

The subsequent objective of CFMIP-2 was to inform improved assessments of climate change cloud feedbacks by providing better tools to support evaluation of clouds simulated by climate models and understanding of cloud-climate feedback processes. CFMIP-2 organized further experiments as part of CMIP5 (Bony et al., 2011; Taylor et al., 2012), introducing seasonally varying SST perturbation experiments for the first time, as well as fixed SST $CO_2$ forcing experiments to examine cloud adjustments. CFMIP-2 also introduced idealized 'aquaplanet' experiments into the CMIP family of experiments. These experiments were motivated by extensive research in the framework of the aqua-planet experiment (Neale and Hoskins, 2000, Blackburn and Hoskins, 2013) and the particular finding, based on a small subset of models, that the global mean cloud feedback of more realistic model configurations could be reproduced, and more easily investigated, using the much simpler aqua-planet configuration (Medeiros et al., 2008). CFMIP-2 proposed the inclusion of the abrupt $CO_2$ quadrupling AOGCM experiment in the core experiment set of CMIP5, based on the approach of Gregory et al., 2004, which subsequently formed the basis for equilibrium climate sensitivity estimates from AOGCMs (Andrews et al., 2012). Additionally CFMIP-2 introduced satellite simulators to CMIP via the CFMIP Observation Simulator Package (COSP, Bodas-Salcedo et al., 2011); not only the ISCCP simulator, but additional simulators to facilitate the quantitative evaluation clouds using a new generation of active radars and lidars in space. CFMIP-2 also introduced into CMIP5 process diagnostics such as temperature and humidity budget tendency terms and high frequency 'cfSites' outputs at 120 locations around the globe. In an effort less directly connected to CMIP, CFMIP organized a joint project with the GEWEX Global Atmospheric System Study (GASS) called CGILS (the CFMIP-GASS Intercomparison of LES and SCMs) to develop cloud feedback intercomparison cases to assess the physical credibility of cloud feedbacks in climate models by comparing Single Column Models (SCM) versions of GCMs with high resolution Large Eddy Simulations (LES) models. CFMIP-2 also developed the CFMIP-OBS data portal and the CFMIP diagnostic codes catalogue. For more details, and for a full list of CFMIP related publications, please refer to the CFMIP web site (http://www.earthsystemcog.org/projects/cfmip).

Studies arising from CFMIP-2 include numerous single and multi-model evaluation studies which use COSP to make quantitative and fair comparisons with a range of satellite products (e.g. Nam et al., 2012; Gregory and Chepfer, 2012; Kay et al., 2012; Franklin et al., 2013; Klein et al., 2013, Lin et al., 2014.). COSP has also enabled studies attributing cloud feedbacks and cloud adjustments to different cloud types (e.g. Zelinka et al., 2013; Zelinka et al., 2014; Tsushima et al., 2015). CFMIP-2 additionally enabled the finding that idealized 'aquaplanet' experiments without land, seasonal cycles or Walker circulations are able to reproduce the essential differences between models' global cloud feedbacks and cloud adjustments in a substantial ensemble of models (Ringer et al., 2014; Medeiros et al., 2015). Process outputs from CFMIP have also been used to develop and test physical mechanisms proposed to explain and constrain inter-model spread in cloud feedbacks in the CMIP5 models (e.g. Webb and Lock, 2013; Sherwood et al., 2014; Brient et al., 2015; Webb et al., 2015a; Nuijens et al., 2015a,b; Dal Gesso at al., 2015). CGILS has demonstrated a consensus in the responses of LES models to climate forcings and identified shortcomings in the physical representations of cloud feedbacks in climate models (e.g.





Blossey et al., 2013; Zhang et al., 2013; Dal Gesso at al., 2015). The CFMIP experiments have additionally formed the basis
for coordinated experiments to explore the impact of cloud radiative effects on the circulation (Stevens et al., 2012; Fermepin
and Bony 2014; Crueger and Stevens 2015; Li et al., 2015; Harrop and Hartmann 2016), the impact of parametrized
convection on cloud feedback (Webb et al., 2015b) and the mechanisms of negative shortwave cloud feedback in mid to high
latitudes (Ceppi et al., 2015). Additionally the CFMIP experiments have, due to their idealized nature, proven useful in a
number of studies not directly related to clouds, but instead analyzing the responses of regional precipitation and circulation
patterns to $CO_2$ forcing and climate change (e.g. Bony et al., 2013; Chadwick et al., 2014; He and Soden 2015; Oueslati et al.,
2016). Studies using CFMIP-2 outputs from CMIP5 remain ongoing and further results are expected to feed into future
assessments of the representation of clouds and cloud feedbacks in climate models.
The primary goal of CFMIP is to inform improved assessments of cloud feedbacks on climate change. However, the CFMIP
approach is increasingly being used to understand other aspects of climate response, such as regional circulation and
precipitation changes, and non-linear changes. This involves bringing climate modelling, observational and process
modelling communities closer together and providing better tools and community support for evaluation of clouds and cloud
feedbacks simulated by climate models and for understanding of the mechanisms underlying them. This is achieved by:
• Coordinating model inter-comparison activities which include experimental design as well as specification of
model output diagnostics to support quantitative evaluation of modelled clouds with observations (e.g. COSP)
and in-situ measurements (e.g. cfSites) as well as process-based investigation of cloud maintenance and
feedback mechanisms (e.g. cfSites, temperature and humidity tendency terms)
• Developing and improving support infrastructure including COSP, CFMIP-OBS and the CFMIP diagnostic
codes catalogue.
• Fostering collaboration with the observational and cloud process modelling communities via annual CFMIP
meetings international funded projects.
CFMIP-3 touches, to differing degrees, on each of the three questions around which CMIP6 is organized (Eyring et al.,
2015). With its focus on cloud feedback, CFMIP-3 is central to CMIP6's attempt to answer the question: `How does the
Earth system respond to forcing?' But as illustrated in the remainder of this document, CFMIP-3 also offers the opportunity
to contribute to the other two guiding questions of CMIP6. Through its strong model evaluation component it stands to help
answer the question: `What are the origins and consequences of systematic model biases?' CFMIP-3 will also help answer the
question: `How can we assess future climate changes given climate variability, climate predictability, and uncertainties in
scenarios?' For example the *amip-piForcing* experiment proposed below will support studies relating cloud variability and
feedbacks on observable timescales to long term cloud feedbacks (Andrews, 2014; Gregory and Andrews, submitted).
The CFMIP experiments proposed for CMIP6, here referred to as CFMIP-3 are outlined below in Section 2. It is anticipated
that CFMIP-3 will eventually be broader than what is described here, for instance including studies with process models, but
for the purposes of this document CFMIP-3 should be considered to be synonymous with the CFMIP contribution to CMIP6.
Section 3 describes the diagnostics outputs proposed for the CFMIP-3, CMIP DECK and CMIP6-Historical experiments. We
provide a summary of the CFMIP-3 contribution to CMIP6 in Section 5.

## 2 CFMIP-3 Experiments

The CFMIP-3 experiments are summarised in Figure 1 and Tables 1 and 2, and are described in detail below. Following the
CMIP6 design protocol, groups of experiments are motivated by science questions and are separated into Tiers 1 and 2
(Eyring et al., 2015). It is a requirement for participation by modelling groups in the CFMIP-3/CMIP6 model
intercomparison that all Tier 1 experiments be performed and published through the ESGF, so as to support CFMIP's Tier 1
science question. Tier 2 experiments are optional, and are associated with additional science questions. Any subset of Tier
2 experiments may be performed. All model output archived by CFMIP/CMIP6 is expected to be made available under the
same terms as CMIP output. Most modelling groups currently release their CMIP data for unrestricted use. Our analysis
plans for the CFMIP-3 experiments are summarised in Appendix A.

## 2.1 CFMIP-3 Tier 1 Experiments

Lead coordinator: Mark Webb

Science Question: What are the physical mechanisms underlying the range of cloud feedbacks and cloud adjustments
predicted by climate models, and which of the cloud responses are the most credible?

Equilibrium climate sensitivity (ECS) can be estimated using an idealized AOGCM experiment such as the *abrupt-4xCO2*
experiment in the CMIP6 DECK, at the same time statistically separating the global mean contributions from climate
feedbacks and adjusted radiative forcing due to $CO_2$ (Gregory et al. 2004, Andrews et al., 2012). However understanding the
physical processes underlying cloud feedbacks and adjustments requires diagnosis in SST forced experiments, which can





resolve cloud feedbacks and adjustments independently from each other and with minimal statistical noise at regional scales,
while faithfully reproducing the inter-model differences in global values from the fully coupled models (Ringer et al., 2014).
(The ability of these AGCM experiments to reproduce the inter-model differences in global cloud feedbacks and adjustments
from coupled models indicates that they do not strongly depend on different ocean model formulations or SST biases). The
CFMIP-2 *amip4xCO2* experiments in CMIP5, which quadrupled $CO_2$ while leaving SSTs at present day values (Bony et al.,
2011), allowed the land/tropospheric adjustment process and the cloud adjustment to $CO_2$ to be examined in this way for the
first time in the multi-model context (Kamae and Watanabe, 2012; Ringer at al., 2014; Kamae et al. 2015) in conjunction
with the CMIP5 *sstClim/sstClim4xCO2* experiments which were based on climatological preindustrial SSTs (Andrews et al.,
2012; Zelinka et al., 2013; Vial et al., 2013). These experiments have additionally formed the basis for more in-depth studies
with individual models (e.g. Wyant et al., 2012; Kamae and Watanabe, 2013; Bretherton et al., 2014, Ogura et al., 2014).
The CFMIP-2/CMIP5 *amip4K* and *amipFuture* SST perturbed atmosphere-only experiments (Bony et al., 2011) have been
used to examine cloud feedbacks in greater detail, often in conjunction with CFMIP process diagnostics (e.g. Brient and
Bony, 2012; Webb and Lock, 2013; Brient and Bony, 2013; Ringer et al., 2014; Bretherton et al., 2014; Lacagnina et al.,
2014; Gordon and Klein, 2014; Chepfer et al., 2014; Sherwood et al., 2014; Medeiros et al., 2015; Brient et al., 2015;
Tsushima et al., 2015; Bellomo and Clement, 2015; Dal Gesso at al., 2015; Webb et al., 2015a, Webb et al., 2015b, Ceppi et
al., 2016). Similarly, these experiments have been used to investigate responses of regional precipitation, circulation and
stability to direct radiative forcing due to increasing $CO_2$ concentrations and/or increases in SST (Bony et al. 2013; Ma and
Xie, 2013; Huang et al., 2013; Widlansky et al., 2013; He et al., 2014; Zhou et al., 2014; Chadwick et al. 2014; Grise and
Polvani, 2014; Kamae et al., 2014; Ceppi et al., 2014; Xie et al. 2015; Qu et al., 2015; Bellomo and Clement, 2015; Shaw and
Voigt, 2015; Kent et al., 2015; Long et al., 2016; Chadwick, 2016).
A more idealized set of fixed SST experiments proposed by CFMIP-2 for CMIP5 (*aquaControl*, *aqua4xCO2*, and *aqua4K*)
based on zonally symmetric, fixed season 'aquaplanet' configurations without land have been shown to reproduce the inter-
model differences in global mean cloud adjustments and feedbacks from realistic experiments surprisingly effectively
(Medeiros et al., 2008; Ringer et al., 2014; Medeiros et al., 2015) as well as many aspects of the zonal mean circulation
response (Medeiros et al., 2015). This indicates that those features of the climate system excluded from these experiments
(i.e. the ocean, land, seasonal cycle, monsoon and Walker circulations) are not central to understanding inter-model
differences in global mean cloud feedbacks and adjustments, and demonstrates the value of aquaplanet experiments for
investigating the origin of such differences, as well as differences in zonally averaged precipitation and circulation and their
responses to climate change (e.g. Stevens et al., 2012; Bony et al., 2013; Brient and Bony, 2013; Kamae and Watanabe 2013;
Oueslati and Bellon, 2013; Fermepin and Bony 2014; Qu et al., 2015; Voigt and Shaw 2015; Harrop and Hartmann, 2015;
Ceppi et al., 2015). The aquaplanet experiments have the benefit not only of being less computationally expensive than
alternative experiments (requiring only 5-10 years to get a robust signal); they are also much more straightforward to analyse,
as their behaviour can mostly be characterized by examining zonal means, avoiding the analysis overhead of compositing
which is generally required in realistic model configurations to isolate the various cloud regimes. Because for the Aqua-
planet simulations it is not possible to tune the models to reproduce a known answer, these (and other idealized) experiments
are particularly effective at highlighting model differences, for instance in the placement of the tropical rain bands, or in the
representation of cloud changes with warming (e.g., Stevens and Bony, 2013).
The CMIP5/CFMIP-2 experiments and diagnostic outputs have thus enabled considerable progress on a number of questions.
However, participation by a larger fraction of modelling groups is desired in CMIP6 to enable a more comprehensive
assessment of the uncertainties across the full multi-model ensemble. Our proposal is therefore to retain the CFMIP-2/CMIP5
experiments (known in CMIP5 as *amip4K*, *amip4xCO2*, *amipFuture*, *aquaControl*, *aqua4xCO2* and *aqua4K*) in Tier 1 for
CFMIP/CMIP6. These are summarised in Table 1 (the names have been changed slightly compared to the CMIP5 equivalents
to fit in with a wider naming convention of CMIP6). The set up for each of these experiments is described below. (For
output requirements from these and other experiments please refer to Section 3).
*amip*: This is a single ensemble member of the CMIP DECK *amip* experiment which contains additional outputs which are
required both for model evaluation using COSP, and for interpretation of feedbacks and adjustments in conjunction with the
*amip-p4K*, *amip-4xCO2*, *amip-future4K* and *amip-m4K* experiments.
*amip-p4K* (formerly *amip4K*): The same as the *amip* DECK experiment, except that SSTs are subject to a uniform warming
of 4K. This warming should be applied to the ice free ocean surface only. Sea ice and SSTs under sea ice remain the same
as in the *amip* DECK experiment.
*amip-future4K* (formerly *amipFuture*): The same as the *amip* DECK experiment, except that a composite SST warming
pattern derived from the CMIP3 coupled models is added to the AMIP SSTs (see Appendix C for details). As with the *amip-
p4K* experiment, the warming pattern should only be applied to the ice free ocean surface, and sea ice and SSTs under sea ice
should remain the same as in the *amip* DECK experiment. The warming pattern should be scaled to ensure that the global
mean SST increase averaged over the ice free oceans is 4K.





*amip-4xCO2 (*formerly *amip4xCO2): *The same as the *amip* experiment within the DECK, except that the $CO_2$ concentration
seen by the radiation scheme is quadrupled. The $CO_2$ seen by the vegetation should be the same as in the *amip* DECK
experiment. This experiment gives an indication of the adjusted radiative forcing due to $CO_2$ quadrupling, including
stratospheric, land surface, tropospheric and cloud adjustments.

The configuration of the *aqua-control, aqua-p4K* and *aqua-4xCO2* experiments are unchanged compared to their equivalents
in CFMIP-2/CMIP5, except that the simulation length has been extended to 10 years to improve the signal to noise ratio.
Further details of their experimental set up are included in Appendix B.

We also propose to use the Tier 1 experiments as the foundation for further experiments planned in the context of the Grand
Challenge on Clouds, Circulation and Climate Sensitivity (Bony et al., 2015). These will include for example sensitivity
experiments to assess the impacts of different physical processes on cloud feedbacks and regional circulation/precipitation
responses and also to test specifically proposed cloud feedback mechanisms (e.g. Webb et al., 2015b, Ceppi et al., 2015).
Additional experiments further idealizing the aquaplanet framework to a non-rotating rotationally symmetric case are also
under development (e.g. Popke et al., 2013). These will be proposed as additional Tier 2 experiments at a future time, or
coordinated by CFMIP outside of CMIP6.

## 2.2 amip minus 4K Experiment (Tier 2)

Lead Coordinators: Mark Webb and Bjorn Stevens

Science Question: Are cloud feedbacks consistent for climate cooling and warming, and if not, why?

There is some evidence to suggest that cloud feedbacks might operate differently in response to cooling rather than warming.
For example, Yoshimori et al., 2009 found a positive shortwave cloud feedback in a $CO_2$ doubling experiment with a
particular GCM, but noted a tendency for it to become weaker or even negative in cooling experiments designed to replicate
the climate of the last glacial maximum. They suggested that this might be related to different displacements of mixed-phase
clouds in the two scenarios. For small enough changes where linearity is a good approximation, one would expect the cloud
response to cooling and warming to be the same, differing only in sign, resulting in an identical cloud feedback expressed per
degree of global temperature change. But for larger perturbations this symmetry of response may no longer hold. A
warming or cooling of the atmosphere of equal magnitude while maintaining relative humidity will for example generate
different changes in absolute humidity, and its horizontal and vertical gradients, which have been linked to cloud feedbacks
(Brient and Bony, 2013; Sherwood et al., 2014), the atmospheric lapse rate and circulation which influences clouds and
depends in part on the absolute humidity (Held and Soden, 2006; Qu et al., 2015) and additionally on extratropical cloud
optical depth feedbacks which may be related to adiabatic cloud liquid water contents (Gordon and Klein, 2014) or phase
changes that depend upon whether a given volume crosses the 0 degree isotherm in the climate change (Ceppi et al. 2015).

The configuration of the *amip-m4K* experiment will be the same as the *amip-p4K* experiment, except that the sea surface
temperatures are uniformly reduced by 4K rather than increased. This cooling should be applied to the ice free ocean surface
only. Sea ice and SSTs under sea ice remain the same as in the *amip* DECK experiment. This experiment will contain
CFMIP COSP and process outputs so as to support the investigation of inconsistent responses of clouds to a cooling vs. a
warming climate in a controlled way through comparison with the *amip-p4K* experiment. This experiment also complements
the abrupt 0.5xCO2 and the -4% solar experiments in that one can identify asymmetries in the warming/cooling response with
and without interactions with the ocean. As such we hope that these experiments will provide useful synergies with
Palaeoclimate Model Intercomparision Project (PMIP).

## 2.3 Atmosphere-only experiments without longwave cloud radiative effects. (Tier 2)

Lead Coordinators: Sandrine Bony and Bjorn Stevens

Science question: How do cloud-radiative effects impact the structure, the strength and the variability of the general
atmospheric circulation in present and future climates?

It is increasingly recognized that clouds, and atmospheric cloud-radiative effects in particular, play a critical role in the
general circulation of the atmosphere and its response to global warming or other perturbations: they have been found to
modulate the structure, the position and shifts of the ITCZ (e.g. Slingo and Slingo 1988; Randall et al., 1989; Sherwood et al
1994; Bergman and Hendon 2000; Hwang and Frierson, 2013; Fermepin and Bony 2014; Voigt et al., 2014; Loeb et al.,
2015; Voigt and Shaw, 2015), the organisation of convection in tropical waves, Madden-Julian Oscillations and other forms
of convective aggregation (e.g. Lee et al., 2001; Lin and Mapes, 2004; Bony and Emanuel, 2005; Zurovac-Jevtic et al., 2006;
Crueger and Stevens, 2015; Muller and Bony, 2015), the extra-tropical circulation and the position of eddy-driven jets





(e.g.Ceppi et al., 2012; Ceppi et al., 2014; Grise and Polvani 2014; Li et al., 2015), and modes of interannual to decadal
climate variability (e.g. Bellomo et al., 2015; Rädel et al., 2016; Yuan et al., 2016). A better assessment of this role would
greatly help to interpret model biases (how much do biases in cloud-radiative properties contribute to biases in the structure
of the ITCZ, in the position and strength of the storm tracks, in the lack of intra-seasonal variability, etc) and to inter-model
differences in simulations of the current climate and in climate change projections (especially changes in regional
precipitation and extreme events). More generally, a better understanding of how clouds couple to the circulation is expected
to improve our ability to answer the four science questions raised by the WCRP Grand Challenge on Clouds, Circulation and
Climate Sensitivity (Bony et al., 2015).

These questions provided the scientific motivation for the Clouds On/Off Klima Intercomparison Experiment (COOKIE)
project proposed by the European consortium EUCLIPSE and CFMIP (Stevens et al., 2012). The COOKIE experiments,
which have been run by four to eight climate models (depending on the experiment), switched off the cloud-radiative effects
(clouds seen by the radiation code -and the radiation code only- were artificially made transparent) in an atmospheric model
forced by prescribed SSTs. By doing so, the atmospheric circulation could feel the lack of cloud-radiative heating within the
atmosphere, but the land surface could also feel the lack of cloud shading, which led to changes in land surface temperatures
and land-sea contrasts. The change in circulation between On and Off experiments resulted from both effects, obscuring to
some degree the mechanisms through which the atmospheric cloud-radiative effects interact with the circulation for given
surface boundary conditions. As the longwave cloud-radiative effects are felt mostly within the troposphere (representing
most of the net atmospheric cloud-radiative heating) while the shortwave effects are felt mostly at the surface (e.g. L'Ecuyer
and McGarragh 2010; Haynes et al., 2013), we could better isolate the role of tropospheric cloud-radiative effects on the
circulation by running atmosphere-only experiments in which clouds are made transparent to radiation only in the longwave.

Therefore we propose in Tier 2 a set of simple experiments similar to the *amip*, *amip-p4K*, *aqua-control* and *aqua-p4K*
experiments within Tier 1, but in which cloud-radiative effects are switched off in the longwave part of the radiation code.
These experiments will be referred to as *amip-lwoff, amip-p4K-lwoff, aqua-control-lwoff* and *aqua-p4K-lwoff*. The analysis of
idealized (aqua-planet) experiments will allow us to assess the robustness of the impacts found in more realistic (AMIP)
configurations. It will also facilitate the interpretation of the results using simple dynamical models or theories, in
collaboration with large-scale dynamicists (e.g. DynVar). The comparison of the inter-model spread of simulations between
the standard and 'lwoff' experiments for present-day and warmer climates will help to identify which aspects of the inter-
model spread depend on the representation of cloud-radiative effects, and which aspects do not, thus better highlighting other
sources of spread.

## 2.4 Abrupt +/-4% Solar Forced AOGCM experiments (Tier 2)

Lead coordinators: Chris Bretherton, Roger Marchand, Bjorn Stevens

Science Question: How do responses in the climate system due to changes in solar forcing differ from changes due to $CO_2$,
and is the response sensitive to the sign of the solar forcing?

While rapid adjustments in clouds and precipitation can easily be separated from conventional feedbacks in SST forced
experiments, such a separation in coupled models is complicated by various issues, including the response of the ocean on
decadal timescales. A number of studies have examined cloud feedbacks in coupled models subject to a solar forcing, which
is generally associated with much smaller global cloud and precipitation adjustment, due to a smaller atmospheric absorption
for a given top of atmosphere forcing (e.g. Lambert and Faull, 2007; Andrews et al., 2010), but the regional cloud and
precipitation changes have yet to be rigorously investigated across models. Solar forcing also differs from greenhouse
forcing through its different fingerprint on the vertical structure of warming (Santer et al., 2013) and small changes in the
radiative heating near the tropopause may project measurably on tropospheric climate (e.g., Butler et al., 2010), for instance
by influencing the baroclinicity in the upper troposphere and thus the storm-tracks (Bony et al., 2015).

A +4% solar experiment *abrupt-solp4p* would be analogous to the *abrupt-4xCO2* experiment but rather than changing $CO_2$ it
would abruptly increase the solar constant by four percent and keep it fixed for 150 years, resulting in a radiative forcing of a
similar magnitude to that due to $CO_2$ quadrupling. This complements the DECK *abrupt-4xCO2* experiment, tests the forcing
feedback framework for analyzing climate change, and would support our understanding of regional responses of the coupled
system with and without $CO_2$ adjustments. The complementary -4% abrupt solar forcing experiment (*abrupt-solm4p*) would
allow the examination of feedback asymmetry under climate cooling, and would also help with the interpretation of model
responses to geo-engineering scenarios and volcanic forcing, and of past climate signals.

## 2.5 nonLinMIP abrupt 2xCO$_2$ and abrupt 0.5xCO$_2$ Experiments (Tier 2)

Lead Coordinator: Peter Good





Science Question: To what extent is regional-scale climate change per $CO_2$ doubling state-dependent (nonlinear); what are
the associated mechanisms; and how does this affect our understanding of climate model uncertainty?

Recent studies with individual, or a small number of climate models, have found substantial nonlinearities in regional-scale
precipitation change (Good et al., 2012; Chadwick and Good, 2013), associated with robust physical mechanisms (Chadwick
and Good, 2013). Significant nonlinearity has also been found in global and regional-scale warming (e.g. Colman and
McAvaney, 2009; Jonko et al., 2013; Good et al., 2015) and ocean heat uptake (Bouttes et al., 2015).

To address this science question we propose two new experiments for Tier 2, abrupt $2xCO_2$ and abrupt $0.5xCO_2$, based on a
proven analysis approach, including traceability of these experiments to transient-forcing simulations (Good et al.,
submitted), to explore global and regional-scale nonlinear responses, highlighting different behaviour under business-as-usual
scenarios, mitigation scenarios and palaeoclimate simulations. Additionally comparisons of the abrupt $2xCO_2$ and abrupt
$4xCO_2$ experiments will help to establish the extent to which the latter accurately estimates the equilibrium climate sensitivity
to CO2 doubling. Additional experiments (Good et al., submitted) may be proposed for Tier 2 in the future, or coordinated
via CFMIP outside of CMIP6. These include 100-year extensions to *abrupt-4xCO2* and *abrupt-2xCO2*; a 1% ramp-down
from the end of the *1pctCO2* experiment; an abrupt step-down to 1xCO2 from year 100 of the *abrupt-4xCO2*. These would
be used to explore longer-timescale responses, quantify nonlinear mechanisms more precisely and understand the reversibility
of climate change.

## 2.6 Feedbacks in AMIP experiments (Tier 2)

Lead Coordinator: Timothy Andrews

Science question: Are climate feedbacks during the 20[th] century different to those acting on long term climate change?

Recent studies have shown significant time variation in climate feedbacks in response to $CO_2$ quadrupling (e.g. Andrews et
al., 2012; Geoffroy et al., 2013; Armour et al., 2013; Andrews et al., 2015). This raises the possibility that feedbacks during
the 20[th] century may be different to those acting on long term change, and hence has the potential to alleviate the apparent
discrepancy between estimates of climate sensitivity from comprehensive climate models and from simple climate models
fitted to observed warming trends (Collins et al., 2013). For example Gregory and Andrews (submitted) found that two
models forced with observed monthly 20[th] century SST and sea-ice variations simulated effective climate sensitivities of
about 2K, whereas these same models forced with patterns of long term SST change simulated effective climate sensitivities
of over 3K and 4K.

The previous CFMIP-2/CMIP5 design was unable to diagnose the time-variation of feedbacks of explicit relevance to the
historical period. To address this we propose an additional experiment called *'amip-piForcing'* (amip pre-industrial forcing)
following the design of Andrews (2014) and Gregory and Andrews (submitted). This experiment is the same as the standard
*amip* run (i.e. using observed monthly updating SSTs and sea-ice), but run for the period 1870-present and with constant pre-
industrial forcings (i.e. all anthropogenic and natural forcing boundary conditions identical to the *piControl* run). Since the
forcing constituents do not change in this experiment it readily allows a simple diagnosis of the simulated atmospheric
feedbacks to observed SST and sea-ice changes, which can then be compared to feedbacks representative of long term change
and climate sensitivity (e.g. from *abrupt-4xCO2* or *amip-p4K*). The experiment has the additional benefit, by differencing
with the standard *amip* run that includes time-varying forcing agents, of providing detailed information on the transient
effective radiative forcing and adjustments in models (Andrews, 2014). This can then be compared to the forcings diagnosed
in RFMIP (who use a pre-industrial climate baseline) to test for any dependence of forcing and adjustments on the climate
state. The experiment therefore complements the alternative approach of diagnosing time-varying feedbacks, which first
requires estimating the forcing and adjustments (e.g. from RFMIP) and removing them from the standard *amip* experiment,
since the approach here extends the time-period of the *amip* simulation and only requires a single experiment (rather than
pairs) which reduces the noise. The inclusion of CFMIP process diagnostics will also enable a deeper understanding of the
factors underlying forcing and feedback differences in the present and future climate.

## 2.7 Time slice experiments for understanding regional climate responses to $CO_2$ (Tier 2)

Lead Coordinators: Robin Chadwick, Hervé Douville and Christopher Skinner

Science questions:
● How do regional climate responses (e.g. of precipitation) in a coupled model arise from the combination of
responses to different aspects of $CO_2$ forcing and sea surface warming (uniform SST warming, patterned SST
warming, sea-ice change, direct $CO_2$ effect, plant physiological effect)?
● Which aspects of forcing/warming are most important for causing inter-model uncertainty in regional climate
projections?



● Can inter-model differences in regional projections be related to underlying structural or resolution differences
between models through improved process understanding, and could this help us to constrain the range of regional
projections?
● What impact do coupled model SST biases have on regional climate projections?
The CFMIP-2/CMIP5 set of idealised amip experiments (e.g. *amip4K*, *amipFuture*) have allowed the contribution of different
aspects of SST warming and increased $CO_2$ concentrations to the projections of fully coupled GCMs to be examined (e.g.
Bony et al., 2013; Chadwick et al., 2014; He and Soden, 2015). However the amip experiments were not designed to replicate
coupled GCM responses on a regional scale, and large discrepancies exist between the two in many regions, particularly
when individual models are examined instead of the ensemble mean (Chadwick, 2016). This is largely due to the choice of
present-day and future SST boundary conditions used in the amip experiments, as well as missing processes such as the plant
physiological response to $CO_2$, rather than the lack of air-sea coupling (Skinner et al., 2012).
We propose a new set of 7 30-year atmosphere-only time slice experiments, and one 36-year amip-style experiment, to
decompose the regional responses of each model's *abrupt-4xCO2* run into separate responses to each aspect of forcing and
warming  (uniform SST warming, pattern SST change, sea-ice change, increased $CO_2$, plant physiological effect). As well as
allowing regional responses in each individual model to be better understood, this set of experiments should prove especially
useful for understanding the causes of model uncertainty in regional climate change.
The experiments are:
1) *piSST* – An AGCM experiment with monthly-varying SSTs, sea-ice, atmospheric constituents and any other necessary
boundary conditions (e.g. vegetation if required) taken from a section of each model's own *piControl* run, using the 30 years
of *piControl* that are parallel to years 111-140 of its *abrupt-4xCO2* run. Note that dynamic vegetation (if included in the
model) should not be turned on in any of the *piSST* set of experiments;
2) *piSST-pxK* – same as *piSST*, but with a global spatially and temporally uniform SST anomaly applied on top of the
monthly-varying *piSST* SSTs. The magnitude of the uniform increase is taken from each model's global, climatological
annual mean SST change between *abrupt-4xCO2* and *piControl* (using the mean of years 111-140 of *abrupt-4xCO2*, and the
parallel 30-year section of *piControl*). Sea-ice is unchanged from *piSST* values;
3) *piSST-4xCO2-rad* – same as *piSST* but $CO_2$ as seen by the radiation scheme is quadrupled;
4) *piSST-4xCO2* – same as *piSST* but with $CO_2$ quadrupled, and this increase is seen by both the radiation scheme and the
plant physiological effect. If a model does not include the plant physiological response to $CO_2$, then *piSST-4xCO2* can be
omitted from the set of *piSST* experiments for that model;
5) *a4SST* – same as *piSST*, but with monthly-varying SSTs taken from years 111-140 of each model's own *abrupt-4xCO2*
experiment instead of from *piControl* (sea ice is unchanged from *piSST*);
6) *a4SSTice* – same as *piSST*, but with monthly-varying SSTs and sea-ice taken from years 111-140 of each model's own
*abrupt-4xCO2* experiment instead of from *piControl*;
7) *a4SST-4xCO2*– same as *piSST*, but with monthly-varying SSTs and sea-ice taken from years 111-140 of each model's own
*abrupt-4xCO2* experiment instead of from *piControl*. $CO_2$ is also quadrupled, and is seen by both the radiation scheme and
the plant physiological effect (if included in the model). *a4SST-4xCO2* is used to establish whether a time slice experiment
can adequately recreate the coupled *abrupt-4xCO2* response in each model, and then forms the basis for a decomposition
using the other experiments.
8) We also propose an additional amip based experiment, *amip-a4SST-4xCO2*: the same as amip, but a patterned SST
anomaly is applied on top of the monthly-varying amip SSTs. This anomaly is a monthly climatology, taken from each
model's own *abrupt-4xCO2* run minus *piControl* (using the mean of years 111-140 of *abrupt-4xCO2*, and the parallel 30-year
section of *piControl*). $CO_2$ is quadrupled, and the increase in $CO_2$ is seen by both the radiation scheme and vegetation.
Comparison of  *amip-a4SST-4xCO2* and *a4SST-4xCO2* should help to illuminate the impact of SST biases on regional
climate responses in each model, and how this contributes to inter-model uncertainty.

## 439 3 CFMIP Recommended Diagnostic Outputs for CMIP experiments

The CFMIP-3 specific diagnostic request is designed to address the following questions: 1) How well do clouds and other
relevant variables simulated by models agree with observations?  2) What physical processes and mechanisms are important
for a credible simulation of clouds, cloud feedbacks and cloud adjustments in climate models?  4) Which models have the
most credible representations of processes relevant to the simulation of clouds?  5) How do clouds and their changes interact
with other elements of the climate system?
The set of diagnostic outputs recommended for CFMIP-3 is based on that from CFMIP-2, with some modifications.   The
request outlined below is in three parts.  The first part describes an updated set of CFMIP process diagnostics (based on those
in CFMIP-2 which are documented at http://cmip-pcmdi.llnl.gov/cmip5/output_req.html) in terms of the various groups of
variables and the experiments in which they are requested.  This set was drawn up by the CFMIP committee and ratified by
the modelling groups following a presentation at the 2014 CFMIP meeting.  The second part describes recommendations for
COSP outputs in the CFMIP-3, CMIP DECK and CMIP6 Historical experiments.  The third part describes additional
diagnostics requested for evaluation of mean diurnal cycle of tropical clouds and radiation.  The summaries below give an
overview of the diagnostic request; however the definitive and detailed specification is documented in the CMIP6 data





request, available at https://earthsystemcog.org/projects/wip/CMIP6DataRequest (Juckes et al., in preparation.) The changes
in the CFMIP-3 diagnostics relative to those requested for CFMIP-2 are additionally motivated and detailed in the CFMIP
CMIP6 proposal document which is available from the CFMIP website.
CMIP mandates that for participation in the CFMIP-3, modelling groups must commit to performing all of the Tier 1
experiments. In recognition that sufficient resources are not available for all groups to prepare all of the CFMIP-3 specific
diagnostics, these diagnostics are considered to be Tier 2, i.e., not compulsory for participation in CFMIP-3. Nonetheless,
these diagnostics are extremely valuable and all groups with the capacity to do so are very strongly encouraged to provide the
additionally requested CFMIP-3 specific diagnostics.
In the case where CFMIP-3 specific outputs are requested in DECK and CMIP6-Historical experiments, and modelling
groups run more than one ensemble member of an experiment, we request that each set of CFMIP-3 specific outputs are
submitted for one ensemble member only. Having different CFMIP variables in different ensemble members is acceptable,
but submitting them all in the same ensemble member is preferable. We request that the modelling groups provide
information on which CFMIP diagnostic sets are submitted in which ensemble members so that this information can be made
available to those who may be analyzing the output. Our analysis plans for the CFMIP diagnostic outputs in the CMIP
DECK, CMIP6 Historical and CFMIP-3 experiments, including details of the CFMIP Diagnostics Code Catalogue are
summarised in Appendix A.

## 3.1 Process outputs

In CFMIP-2, instantaneous high frequency 'cfSites' outputs were requested for 120 locations in the *amip, amip4K,*
*amipFuture* and *amip4xCO2* experiments, and for 73 locations along the Greenwich meridian in the aquaplanet experiments,
to support understanding and evaluation of clouds and their interactions with convection and other processes. The 120
locations include the locations of instrumented sites (ARM and CloudNet stations, Dome C, etc), the transect associated with
the GCSS Pacific Cross-section Intercomparison (GPCI), past field campaigns (DYCOMS-II, NARVAL, HOPE, VOCALS,
ASTEX and AMMA transects, TOGA-COARE, RICO, etc) and a number of climate regimes that contribute substantially to
the inter-model spread of cloud feedbacks in climate change (Webb et al., 2015a). These outputs have so far been used to
evaluate the models with in-situ measurements (e.g. Nuijens et al., 2015a, Nuijens et al., 2015b, Neggers et al., 2015), to
investigate the diurnal cycle of cloud feedbacks (Webb et al., 2015a) and to compare cloud feedbacks in climate models with
Single Column Models and LES outputs from CGILS (Dal Gesso at al., 2015). We have added St. Helena to the list of
locations in light of upcoming field work, increasing the total number of locations to 121 for CFMIP-3. A text file containing
the list of locations is available in the Supplementary Information and on the CFMIP website; these are also presented
graphically in Figure 2.
For CFMIP-3 we have dispensed with the cfSites outputs in the aquaplanet experiments and in *amip-future4K*. cfSites outputs
are now requested for one ensemble member of the *amip* DECK experiment, and the *amip-p4K* and *amip-4xCO2*
experiments. Outputs should be provided for the full duration of each experiment. The sampling interval should be the
integer multiple of the model time step that is nearest to 30 minutes and divides into 60 minutes with no remainder: e.g. 30
minutes for a 30, 15 or 10 minute time step or 20 minutes for a 20 minute time step. Outputs should be instantaneous (i.e. not
time means) and from nearest grid box (i.e. no spatial interpolation).
The cfSites outputs from CFMIP-3 provide instantaneous outputs of a range of quantities (including temperature and
humidity tendency terms) in experiments which can be used to evaluate the present day relationships of clouds to cloud
controlling factors using in situ measurements, and at the same time explore how these relationships affect cloud feedbacks
and cloud adjustments. An increasing wealth of observational data with which to evaluate the models using these outputs is
available or in the planning stage, for example from the Barbados Cloud Observatory (Stevens et al., 2015) the ARM
Program (e.g. Wood et al., 2015; Marchand et al., 2015) or within the German national project on high-definition clouds and
precipitation for climate-prediction, HD(CP)$^2$, inclusive of its observational prototype experiment (HOPE), and which has
collected observations over Germany following conventions adopted for CMIP (Andrea Lammert, personal communication).
CFMIP-2 also requested cloud, temperature and humidity tendency terms from convection, radiation, dynamics etc. in the
*amip*, *amip4K, amipFuture and amip4xCO2, aquaControl, aqua4xCO2* and *aqua4K* experiments, as global monthly mean
outputs and high frequency outputs at fixed locations (Bony et al., 2011). Upward and downward radiative fluxes on model
levels were also requested in these experiments, and for instantaneous $CO_2$ quadrupling in the *amip* experiment only.
Temperature and humidity tendency terms in particular have been shown to be useful for understanding the roles of different
parts of the model physics in cloud feedbacks and adjustments (Kamae and Watanabe 2012; Williams et al., 2013; Webb and
Lock 2013; Demoto et al., 2013; Sherwood et al., 2014; Ogura et al., 2014; Brient et al., 2015) as well as in understanding
clouds and circulation in the present climate (e.g. Oueslati and Bellon, 2013; Xavier et al., 2015). They have also been used to
understand regional warming patterns such as polar amplification in coupled models (e.g. Yoshimori et al., 2014).
In CFMIP-3 we have dispensed with the cloud tendency terms, improved the definitions of the temperature and humidity
tendency terms, and added some additional terms such as clear-sky radiative heating rates to more precisely quantify the





contributions of different processes to the temperature and humidity budget changes underlying cloud feedbacks and
adjustments. A shortcoming of the CMIP5 protocol was that we were unable to interpret the physical feedback mechanisms
in coupled model experiments due to a lack of process diagnostics. For this reason in CMIP6 we are requesting these budget
terms in the DECK *abrupt-4xCO2* experiment and the pre-industrial control as well as one ensemble member of the *amip*
DECK experiment, and all of the CFMIP-3 experiments listed in Sections 2.1-2.6.
Clustering approaches (e.g., Jakob and Tselioudis, 2003) are now commonly used for assessing the contributions of different
cloud regimes (e.g. stratocumulus, trade cumulus, frontal clouds, etc) to present day biases in cloud simulations and to inter-
model differences in cloud feedbacks (e.g. Williams and Webb 2009, Tsushima et al., 2013, Tsushima et al., 2015). We have
also added some additional daily 2D fields to the standard package of CFMIP daily outputs to allow further investigation of
feedbacks between clouds and aerosols associated with the changing hydrological cycle (aerosol loadings and cloud top
effective radii/number concentrations) and a clearer diagnosis of the roles of convective and stratiform clouds (convective vs.
stratiform ice and condensed water paths and cloud top effective radii/number concentrations).

## 3.2 COSP outputs

This section motivates and summarizes the COSP outputs requested from the DECK, and CMIP6 historical and CFMIP-3
experiments as well as a corresponding set of observations.
There is no unique definition of clouds or cloud types, neither in models nor in observations. Therefore, to compare models
with observations, and even to compare models with each other, it is necessary to use a consistent definition of clouds
between the model and the satellite product in question (i.e., be "definition-aware"). Further complicating matters - climate
model grid boxes (typically 1 degree) are much larger than the scales over which many satellite observations are made
(typically <10 km). As a result, one must downscale the climate model cloud properties to the observation scale (i.e., be
"scale-aware"). The CFMIP Observation Simulator Package (COSP) enables definition-aware and scale-aware comparisons
between models and multiple sets of observations by producing cloud diagnostics from model simulations that are
quantitatively comparable to a variety of satellite products from ISCCP, CloudSat, CALIPSO, MODIS, MISR and Parasol
(Bodas-Salcedo et al., 2011). COSP enables a more quantitative comparison of model outputs with satellite cloud products,
which often sub-sample low level clouds in the presence of high level clouds due to the effects of cloud overlap and
attenuation (e.g. Yu et al., 1996). COSP also provides histograms of various cloud properties as a function of height or
pressure which are directly comparable with satellite products and cannot be calculated correctly from time mean model
outputs. The multiple simulators within COSP allow a multi-faceted evaluation of clouds in models whereby the strengths
and weaknesses of different satellite products may be considered together.
COSP is increasingly being used not only for model intercomparison activities but as part of the model development and
evaluation process by modelling groups (e.g. Marchand et al., 2009; Zhang et al., 2010; Kay et al., 2012; Franklin et al.,
2013; Lacagnina and Selten, 2014; Nam et al., 2014; Williams et al., 2015, Konsta et al., 2015). Many of the standard
monthly and daily COSP outputs have been shown to be valuable in the CMIP5 experiments, not only for cloud evaluation,
allowing a detailed evaluation clouds and precipitation, and their interaction with radiation (e.g. Nam et al., 2012; Cesana and
Chepfer, 2012; Kay et al. 2012; Klein et al., 2013; Tsushima et al., 2013; Gordon and Klein, 2014; Lin et al., 2014; Bodas-
Salcedo et al., 2014; Bellomo and Clement, 2015), but also in quantifying the contributions of different cloud types to cloud
feedbacks and forcing adjustments in climate change experiments (e.g. Zelinka et al., 2013; Zelinka et al., 2014; Chepfer et
al., 2014; Tsushima et al., 2015). For a full list of studies that use COSP diagnostics for model evaluation and feedback
analysis please refer to the 'CFMIP publications' section of the CFMIP website.
Here we will give only a brief overview of the COSP request; readers interested in the complete details of the data request are
referred to the Earth System CoG website (https://earthsystemcog.org/projects/wip/CMIP6DataRequest).
The COSP data request for the CMIP DECK and CMIP6 has been designed to span model evaluation across different space
and time scales. Monthly-mean diagnostics allow for the evaluation and intercomparison of large-scale distributions of cloud
properties and their interaction with radiation. High-frequency model outputs (daily, 3-hourly) are aimed at a process-oriented
evaluation (e.g. Bodas-Salcedo et al., 2012) and offer the opportunity of exploiting the synergy between multiple instruments
(e.g. Konsta et al., 2015). Recent observational developments have improved our capability to retrieve cloud radiative
properties. In particular, new methodologies for cloud phase identification are available for CALIPSO and MODIS, and
COSP has been enhanced to provide diagnostics that are compatible with these new observational datasets (Cesana and
Chepfer, 2013). These new diagnostics will help elucidate some open questions regarding the role of cloud phase in model
biases (Ceppi et al., 2016; Bodas-Salcedo et al., in press).
Within CFMIP-3 COSP output is requested from six simulators as follows:
• ISCCP: pseudo-retrievals of cloud top pressure (CTP) and cloud optical thickness (tau) (Klein and Jakob 1999;
Webb et al., 2001).
• CloudSat: a forward model for radar reflectivity as a function of height (Haynes et al., 2007).





- CALIPSO (Chepfer et al., 2008; Cesana and Chepfer, 2013): forward model for lidar scattering ratio as function of height, and cloud phase retrieval.
- MODIS: pseudo-retrievals of CTP, effective particle size and tau as function of phase (Pincus et al., 2012).
- MISR: pseudo-retrievals of cloud top height (CTH) and tau (Marchand and Ackerman, 2010).
- PARASOL: simple forward model of mono-directional reflectance (Konsta et al., 2015).

The main difference to CFMIP-2 is that output is requested from a greater number of simulators and longer periods of simulated time. MISR provides more accurate retrievals of cloud-top-height for low-level and mid-level clouds, and more reliable discrimination of mid-level clouds from other clouds, while MODIS provides better retrievals of high-level clouds. ISCCP and MISR histograms can be combined to separate optically-thin high-level clouds into multi-layer and single-layer categories (Marchand et al. 2010). Aerosol schemes are becoming more complex, with more elaborate representations of cloud-aerosol interactions. This makes the evaluation of the phase partitioning an important aspect of model evaluation, and height-resolved partitioning estimates from the CALIPSO simulator are included in the COSP request. Cloud phase and particle size estimates from the MODIS simulator were not available in CFMIP-2 but may prove a useful complement to investigate cloud-aerosol interactions by virtue of greater geographic sampling and longer time records. Many of the COSP diagnostics are now requested for the entire lengths of the DECK, CMIP6 Historical and CFMIP-3 experiments to support the quantification and interpretation of cloud feedbacks and cloud adjustments in a broader context. The new inclusion in this COSP request of a long time series of three-dimensional cloud fractions will facilitate the comparison of cloud trends with the observational record (Chepfer et al., 2014). More details of all the changes with respect to CFMIP-2 can be found in the proposal of the CMIP6-Endorsed MIPs, available from the CMIP6 web site (http://www.wcrp-climate.org/wgcm-cmip/wgcm-cmip6).

The COSP output is in six variable groups:

1) CFMIP-cfMon-sim: monthly means of ISCCP 2D diagnostics (cloud fraction, cloud albedo, and cloud top pressure), ISCCP CTP-tau histogram, and CALIPSO 2D and 3D cloud fractions.
2) CMIP5-cfDay-2d: daily means of ISCCP and CALIPSO 2D diagnostics, and PARASOL reflectances.
3) cfDay-3d: daily means of ISCCP and CALIPSO 3D diagnostics.
4) CFMIP-cfMonExtra: monthly means of CloudSat reflectivity and CALIPSO scattering ratio histograms as function of height, CALIPSO 3D cloud fractions by phase, MODIS 2D cloud fractions, MODIS CTP-tau histogram and size-tau histograms by phase, MISR CTH-tau histograms, and PARASOL reflectances.
5) CFMIP-cfDayExtra: daily means of CALIPSO total cloud fraction, MODIS CTP-tau histogram and size-tau histograms by phase, and PARASOL reflectances.
6) CFMIP-cf3hrSim: 3-hourly instantaneous diagnostics of ISCCP CTP-tau histograms, MISR CTH-tau histograms, MODIS CTP-tau histogram and size-tau histograms by phase, CALIPSO 2D and 3D cloud fractions, CloudSat reflectivity and CALIPSO scattering ratio histograms as function of height, and PARASOL reflectances.

The variable groups CFMIP-cfMon-sim and CMIP5-cfDay-2d are requested for all years in the *amip* experiment performed as part of the DECK and the CMIP6-Historical experiments, and for 140 years the *piControl*, *1pctCO2*, and *abrupt-4xCO2*. These are requested for one ensemble member only from these experiments. They are also requested in all of the CFMIP experiments listed in Sections 2.1-2.6 above. cfDay-3d is requested in one ensemble member of the DECK amip experiment and in the CFMIP *amip-p4K* and amip-4xCO2 experiments. CFMIP-cfMonExtra and CFMIP-cfDayExtra are requested for all years of one ensemble member of the *amip* DECK experiment, and CFMIP-cf3hrSim for the year 2008 only.

COSP 1.4, available via the CFMIP website (https://www.earthsystemcog.org/projects/cfmip), is the official version to be used for CMIP6. This is a stable release that was made available well in advance of CMIP6 at the request of the modelling groups. Version 2 of COSP is under active development. At the time of writing, COSP 2 is in beta testing and does not have a stable release, and so is not currently permitted for production of CMIP6 data. COSP-2 may be permitted for use in CMIP6 along with COSP 1.4 in the future; if and when this happens details will be posted on the CFMIP website.

The CFMIP community has developed a set of observational datasets available via the CFMIP-OBS web site (http://climserv.ipsl.polytechnique.fr/cfmip-obs/) that are defined consistently with the COSP diagnostics and the CFMIP data request in terms of vertical grids and time averaging periods. These are mostly reported as monthly means although some are reported at higher temporal resolution for process oriented model evaluations (e.g. Konsta et al., 2012). Table 3 summarizes the datasets relevant to the COSP CMIP6 data request. Some of the CFMIP-OBS datasets listed in Table 3 (CALIPSO, CloudSat, ISCCP, PARASOL) are also available from the ESGF as part of the obs4MIPs project (Teixeira et al., 2014).

## 3.3 Monthly Mean Diurnal Cycle Outputs

Climate models have difficulties representing the diurnal cycle of convective clouds over land (Yang and Slingo, 2001; Stratton and Stirling, 2011), but its evaluation is not possible with sun-synchronous satellites. Geostationary satellites provide high-frequency sampling that can be used to evaluate model biases in the diurnal cycle of clouds and radiation (albeit


over a limited area). The Geostationary Earth Radiation Budget instrument (GERB; Harries et al., 2005) measures the TOA
radiation budget from a geostationary orbit at 0E at 15 minute frequency, which provides a unique view of tropical
convection over Africa. The variable group *cf1hrClimMon* requests monthly mean diurnal cycle of TOA radiative fluxes (all-
sky and clear sky) for the entire length of the *amip* DECK experiment. The radiative fluxes are hourly UTC means. The
'average day' for each month of the simulation is then constructed by averaging each UTC hourly mean over the entire
month. These diagnostics will be directly comparable with GERB measurements.

## 4. Summary

The primary goal of CFMIP is to inform improved assessments of cloud feedbacks on climate change. This involves bringing
climate modelling, observational and process modelling communities closer together and providing better tools and
community support for understanding and evaluation of clouds and cloud feedbacks simulated by climate models. CFMIP
supports ongoing coordinated model inter-comparison activities by recommending  experiments and model output
diagnostics for CMIP, designed to support the understanding and evaluation of cloud processes and cloud feedbacks in
models. The CFMIP approach is also increasingly being used to understand other aspects of climate change, such as
circulation, regional-scale precipitation and non-linear changes. CFMIP proposes a number of experiments and model outputs
for CMIP6, building on and extending those which were part of CMIP5.
A compact set of Tier 1 experiments are proposed address the question: "1) What are the physical mechanisms underlying the
range of cloud feedbacks and cloud adjustments predicted by climate models, and which models have the most credible cloud
feedbacks?"  The Tier 1 experiments (*amip-p4K, amip-4xCO2, amip-future4K, aqua-control, aqua-4xCO2* and *aqua-p4K*)
retain the idealized experimental hierarchy of the CFMIP-2/CMIP5 experiments while building on the DECK AMIP
experiment. A number of Tier 2 experiments are proposed to address additional science questions. An amip uniform minus
4K experiment is proposed to address the question "2) Are cloud feedbacks consistent for climate cooling and warming, and
if not, why?" Atmosphere-only experiments with clouds made transparent to longwave radiation address the question "3)
How do cloud-radiative effects impact the structure, the strength and the variability of the general atmospheric circulation in
present and future climates?" Abrupt +/-4% Solar Forced AOGCM experiments are proposed for the question "4) How do
responses in the climate system due to changes in solar forcing differ from changes due to $CO_2$, and is the response sensitive
to the sign of the solar forcing?"  abrupt $2xCO_2$ and abrupt $0.5xCO_2$ experiments are proposed to address the question "5) To
what extent is regional-scale climate change per $CO_2$ doubling state-dependent (nonlinear), and why?" Other experiments and
questions proposed include: AMIP with preindustrial forcing "6) Are climate feedbacks during the 20[th] century different to
those acting on long term climate change and climate sensitivity?"; Time slice experiments forced with SSTs from
preindustrial and *abrupt-4xCO2* simulations "7) How do regional climate responses (of e.g. precipitation) in a coupled model
arise from the combination of responses to different aspects of $CO_2$ forcing and warming (uniform SST warming, pattern SST
warming, direct CO2 effect, plant physiological effect, sea-ice change)?"
The CFMIP experiments in CMIP6 will continue to include outputs from the CFMIP Observational Simulator Package
(COSP) to support robust scale-aware and definition-aware evaluation of modelled clouds with observations and to relate
cloud feedbacks to observed quantities. COSP outputs are also proposed for inclusion in the DECK and CMIP6 Historical
experiments. Process diagnostics including 'cfSites' high frequency outputs at selected locations and temperature and
humidity budget terms from radiation, convection, dynamics, etc. are also retained from CMIP5. These will help to address
the following questions:  1) How well do clouds and other relevant variables simulated by models agree with observations?
2) What physical processes and mechanisms are important for a credible simulation of clouds, cloud feedbacks and cloud
adjustments in climate models? 4) Which models have the most credible representations of processes relevant to the
simulation of clouds?  5) How do clouds and their changes interact with other elements of the climate system?
By continuing the CFMIP experiments and diagnostic outputs within CMIP6 we hope to apply the well established aspects of
the CFMIP approach to a larger number of climate models.   Additionally we have proposed new experiments to investigate a
broader range of questions relating to the Grand Challenge on Clouds, Circulation and Climate Sensitivity. We hope that the
modelling community will participate fully in CFMIP via CMIP6 so as to maximize the relevance of our findings to future
assessments of climate change.

## Code and Data Availability

COSP is published under and open source license via GitHub (please see the CFMIP website for details). The model output
from the DECK, CMIP6 historical and CFMIP-3 simulations described in this paper will be distributed through the Earth
System Grid Federation (ESGF) with digital object identifiers (DOIs) assigned. As in CMIP5, the model output will be freely
accessible through data portals after registration. In order to document CMIP6's scientific impact and enable ongoing support
of CMIP, users are obligated to acknowledge CMIP6, the participating modelling groups, and the ESGF centres (see details
on the CMIP Panel website at http://www.wcrp-climate.org/index.php/wgcm-cmip/about-cmip). Further information about
the infrastructure supporting CMIP6, the metadata describing the model output, and the terms governing its use are provided





by the WGCM Infrastructure Panel (WIP) in their invited contribution to this Special Issue. Along with the data itself, the
provenance of the data will be recorded, and DOIs will be assigned to collections of output so that they can be appropriately
cited. This information will be made readily available so that published research results can be verified and credit can be
given to the modelling groups providing the data. The WIP is coordinating and encouraging the development of the
infrastructure needed to archive and deliver this information. In order to run the experiments, datasets for natural and
anthropogenic forcings are required. These forcing datasets are described in separate invited contributions to this Special
Issue. The forcing datasets will be made available through the ESGF with version control and DOIs assigned.
Acknowledgements: We thank the modelling groups and wider CFMIP community for reviewing and supporting the CFMIP
contribution to CMIP6, the CMIP Panel for their coordination of CMIP6, the WGCM Infrastructure Panel (WIP) overseeing
the CMIP6 infrastructure, and Martin Juckes for taking the lead in preparing the CMIP6 data request. We are also grateful to
Robert Pincus and Yuying Zhang  for their contributions to COSP and to CFMIP-OBS,  to Dustin Swales for his
development work for COSP-2, and to Gregory Cesana and Mathieu Reverdy for their contributions to CFMIP-OBS.  We are
grateful to Brian Soden for producing the CMIP3 composite pattern dataset used for the CMIP5 *amipFuture* and CMIP6
*amip-future4K* experiments, and to PMIP representatives Pascale Braconnot, Masa Kageyama, and Masakazu Yoshimori for
discussions relating to the amip-m4K experiment. The efforts of S. A. Klein are supported by the Regional and Global
Climate Modeling program of the United States Department of Energy's Office of Science and were performed under the
auspices of the U. S. Department of Energy by Lawrence Livermore National Laboratory under Contract
DEAC5207NA27344.  Met Office Hadley Centre authors are supported by the Joint DECC/Defra Met Office Hadley Centre
Climate Programme (GA01101).

**Appendix A: Analysis Plan and CFMIP Diagnostic Codes Catalogue**

CFMIP-2 analysis activities are ongoing and the CFMIP community is ready to analyse CFMIP-3 data at any time. We would
like modelling groups to perform the proposed CFMIP-3 experiments at the same time or shortly after their DECK and
CMIP6 Historical experiments.  Subsequent CFMIP experiments which are not included in CMIP6 will build on the proposed
DECK and CMIP6/CFMIP experiments and some will start as soon as CMIP6 DECK experiments start to become available.
We envisage a succession of CFMIP related intercomparisons addressing different questions arising from the Grand
Challenge spanning the duration of CMIP6.
We plan to scientifically analyze, evaluate and exploit the proposed experiments and diagnostic outputs, and have identified
leads within CFMIP for different aspects of this activity. An overview of the proposed evaluation/analysis of the CMIP
DECK, CMIP6 Historical and CFMIP CMIP6 experiments follows:
CFMIP will continue to exploit the CMIP DECK and CMIP6 experiments to understand and evaluate cloud processes and
cloud feedbacks in climate models.  The wide range of analysis activities described above in the context of CFMIP-2 will be
continued in CFMIP-3 using the CMIP DECK and CMIP6 experiments, allowing the techniques developed in CFMIP-2 to
applied to an expanding number of models, including the new generation of models currently under development. These
activities will include evaluation of clouds using additional simulators, investigation of cloud processes and cloud
feedback/adjustment mechanisms using process outputs (cfSites, tendency terms, etc). The inclusion of COSP and budget
tendency terms in additional DECK experiments (e.g. *abrupt-4xCO2*) will enable the CFMIP approach to be applied to a
wider range of experimental configurations. Lead coordinator: Mark Webb.
Analysis of the +/-4% solar forcing runs will include an evaluation of both rapid adjustments and longer-term responses on
global and regional top-of-atmosphere radiative fluxes, cloud types (using ISCCP and other COSP simulators) and
precipitation characteristics, as well as comparison of these responses with responses in DECK *abrupt-4xCO2* experiments.
GeoMIP and SolarMIP have expressed a strong interest in these CFMIP experiments and joint analysis of these CFMIP
experiments with GeoMIP and SolarMIP experiments is anticipated, specifically with the goal of determining to what degree
results from abrupt solar forcing only experiments and abrupt $CO_2$ only experiments can be used to predict what happens
when both forcing are applied simultaneously, as done in the GeoMIP experiments. Lead coordinators: Chris Bretherton,
Roger Marchand and Bjorn Stevens.
Analysis of nonlinear climate processes is discussed in detail by Good et al., 2016.  This includes a method for validating
traceability of abrupt $CO_2$ experiments to transient simulations, which is also recommended as a standard test of the DECK
abrupt-4xCO2 experiment.  Analysis will primarily involve comparing the *abrupt-4xCO2*, *abrupt-2xCO2* and *abrupt-
0p5xCO2* experiments over the same timescale. Lead coordinator: Peter Good.
Analysis of *amip-piForcing* has already been performed in detail for two models in Andrews, 2014 and Gregory and
Andrews (submitted).  We propose to use this as a starting point for a multi-model analysis. Lead coordinator: Timothy
Andrews.



An overview analysis of regional responses and model uncertainty in the piSST set of experiments will be carried out by the
coordinators, in collaboration with members of contributing modelling groups. We anticipate that further detailed analysis on
the processes at work in different regions will be carried out by a variety of research groups with interest and expertise in a
particular region: for example a set of similar experiments has previously been used to examine the climate response of the
West African monsoon in CCSM3 (Skinner et al., 2012). The piSST set of experiments have already been successfully run
using the Met Office, NCAR and CNRM CMIP5 models. Lead Coordinators: Robin Chadwick, Hervé Douville and
Christopher Skinner.
The analysis of the COOKIE experiments will be reviewed by the coordinators in collaboration with members of the
contributing modelling groups. The role of longwave atmospheric cloud-radiative effects in large-scale circulations, regional
precipitation patterns and the organisation of tropical convection will be investigated in the current climate and in climate
change, with the aim of highlighting both robust effects and sources of uncertainties in the model responses. Lead
coordinators: Sandrine Bony and Bjorn Stevens.
When analyzed together with the *amip-p4K* experiment, the *amip-m4K* experiment allows the CFMIP process diagnostics to
be used to understand for asymmetries in the climate response to warming and cooling which have been noted in PMIP
experiments. These might arise from cloud phase responses in middle- and high-latitude clouds or from the adiabatic cloud
liquid water path response feedback which is important over land regions and which would be expected to be weaker with
cooling because of the non-linearity in the Clausius-Clapeyron relation. Lead coordinators: Mark Webb and Bjorn Stevens.
The COSP data request for the *amip* DECK experiment will allow a comprehensive multi-model evaluation of clouds and
radiation, following on from CMIP5 studies (e.g. Klein et al., 2013; Bodas-Salcedo et al., 2014). The COSP data request for
the other experiments (e.g. *amip-p4K*, *abrupt-4xCO2*, etc.) permits evaluation of cloud feedbacks and adjustments by cloud
type (Zelinka et al., 2013, Tsushima et al., 2015) or cloud trends (Chepfer et al., 2014). New COSP diagnostics have been
used in single-model analyses: cloud phase diagnostics (Cesana and Chepfer, 2013); MISR simulator outputs to evaluate
cloud fraction and multilayer clouds (Marchand and Ackerman, 2010); CALIPSO vertical distribution of cloud fraction for
the study of cloud trends (Chepfer et al., 2014). These studies will be used as starting points for multi-model analyses. The
COSP PMC co-chairs will coordinate and encourage the exploitation of these resources. Lead coordinators: Alejandro Bodas-
Salcedo and Steve Klein.
Analysis of output from CFMIP and CMIP6 experiments will also be facilitated by sharing of diagnostic codes via the
CFMIP Diagnostics Code Catalogue (accessible via the CFMIP website http://www.earthsystemcog.org/projects/cfmip/).
This is a catalogue of programs written by various members of the CFMIP community, implementing a number of diagnostic
approaches from published studies. These include daily cloud clustering evaluation metrics based on ISCCP and ISCCP
simulator outputs (Williams and Webb, 2009, Tsushima et al., 2013), error metrics for total cloud amount, longwave and
shortwave cloud properties (Klein et al., 2013), process oriented evaluation of clouds using A-train instantaneous
observations (Konsta et al., 2012), quality control and low-cloud diagnostics (Nam et al., 2012; Nam and Quaas, 2012),
sensitivity of low cloud cover to estimated inversion strength and SST (Qu et al., 2013) and cloud radiative kernels (Zelinka
et al., 2012). Any codes which implement diagnostics which are relevant to analysing clouds, circulation and climate
sensitivity in models and which are documented in peer reviewed studies are eligible for inclusion in the catalogue, and we
welcome additional contributions to further support community analysis of CMIP6 outputs.
**APPENDIX B: Aquaplanet Experimental Design**
Aquaplanets are Earth-like planets with completely water-covered surfaces. They are often used as idealized configurations
of atmospheric GCMs, and in this context the usual convention is that landmasses and topography are removed. Although
many flavours of aquaplanet configurations exist, another convention is to retain as much of the atmospheric model's
formulation as possible. That is, the numerical grid, dynamical core, and parameterized physics are all used just as in realistic
climate simulations.
The Tier 1 aquaplanet experiments follow the same experimental design as CMIP5/CFMIP-2 (Medeiros et al., 2015). Those,
in turn, were closely related to previous aquaplanet descriptions. In particular, the control configuration closely follows the
AquaPlanet Experiment protocol (Blackburn and Hoskins, 2013) using a prescribed SST pattern described by Neale and
Hoskins (2000). Two additional runs parallel the CFMIP-2 *amip4K* and *amip4xCO2* experiments: a uniform 4K warming and
a quadrupling of atmospheric $CO_2$.
Here we provide the detailed experimental protocol for the three aquaplanet simulations that are part of Tier 1. We note again
that these follow the APE protocol and CMIP5/CFMIP-2, and therefore largely mirror previous descriptions in Blackburn and
Hoskins (2013), Williamson et al. (2012), and Medeiros et al. (2015).





Orbital parameters are set to perpetual equinox conditions. This is usually achieved by setting eccentricity and obliquity to
zero to define a circular orbit and insolation independent of calendar. The diurnal cycle is retained. Insolation is based on a
non-varying solar constant of 1365 W m$^{-2}$.

The SST is non-varying and zonally uniform. The longitudinal variation is specified using the "Qobs" SST pattern from
Neale and Hoskins (2000), given by:
$$T(\varphi) = \begin{cases} \frac{1}{2}(2 - sin^4\phi - sin^2\phi)\delta T + T_{min}, \text{if } |\varphi| < \frac{\pi}{3} \\ 0, \text{otherwise} \end{cases} \quad \text{(B1)}$$

where $\varphi$ is latitude, $\phi = \frac{\pi}{2}\frac{\varphi}{\varphi_{max}}$, $\varphi_{max} = \frac{\pi}{3}$, $\delta T = T_{max} - T_{min}$, $T_{max} = 27°C$, and $T_{min} = 0°C$.

Because results are sensitive to the specification of the SSTs, groups that use a prognostic equation for the surface skin
temperature are asked to set this skin temperature to the specified SST. No sea ice is prescribed, so the surface temperature is
spatially uniform at $0°C$ poleward of $60°$ for the control simulation.

Radiatively active trace gases are well-mixed with mixing ratios following the AMIP II recommendations: $CO_2$: 348 ppmv;
$CH_4$: 1650 ppbv; $N_2O$: 306 ppbv; Halocarbon yield of approximately 0.24 W m$^{-2}$ radiative forcing. The ozone distribution is
the same as used in APE and CFMIP2/CMIP5, and is derived from the climatology used in AMIP II (Gates et al., 1999), and
is constant in time and symmetric about the equator. This ozone distribution is provided as a netCDF file which is archived
on the Earth System Grid and available via the DOI http://dx.doi.org/10.5065/D61834Q6 (and also available via the CFMIP
website).

Aerosols are removed to the extent possible to remove aerosol-radiation interaction (aka direct effects) and aerosol-cloud
interaction (aka indirect effects). No external surface emissions are to be prescribed. Models requiring aerosol for cloud
condensation should use a constant oceanic climatology that is symmetric about the equator and zonally. Alternatively,
models with the capability should set the cloud droplet and crystal numbers to $100*10^6$ m$^{-3}$ and $0.1*10^6$ m$^{-3}$, respectively (as
in Medeiros et al., 2016).

As in APE, it is recommended that the atmospheric dry mass be adjusted to yield a global mean of 101080 Pa. It is also
recommended to adopt the APE recommended values for geophysical constants, as listed in Table 2 of Williamson et al.
836 (2012).


The aqua-4K experiment follows the above protocol, but with SST derived by adding 4K to Eq. B1.

The aqua-4xCO2 experiment replaces the $CO_2$ mixing ratio with 1392 ppmv. The SST is unchanged from the control
simulation (Eq. B1).

Model runs should be 10 years. We recommend discarding the initial spin up period of a few months.


**APPENDIX C: SST Pattern for CFMIP *amip-future4K*/*amipFuture* experiments**

The *amip-future4K* (formerly *amipFuture*) experiment is the same as the *amip* DECK experiment, except that the SSTs are
subject to a composite SST warming pattern derived from the CMIP3 coupled models. The patterned SST forcing dataset is
available in a netcdf file called cfmip2_4k_patterned_sst_forcing.vn1.0.nc which is available in the supplementary
information for this paper, and via the CFMIP website. This is a normalised multi-model ensemble mean of the ocean
surface temperature response pattern (the change in ocean surface temperature (TOS) between years 0-20 and 140-160, the
time of CO2 quadrupling in the 1% runs) from thirteen CMIP3 AOGCMs (cccma, cnrm, gfdlcm20, gfdlcm21, gisser,
inmcm3, ipsl, miroc-medres, miub, mpi, mri, ncar-ccsm3, and ncar-pcm1.) Before computing the multi-model ensemble
mean, each model's TOS response was divided by its global mean and multiplied by 4. This guarantees that the pattern
information from all models is weighted equally and the global mean SST forcing is the same as in the uniform +4K
experiment.




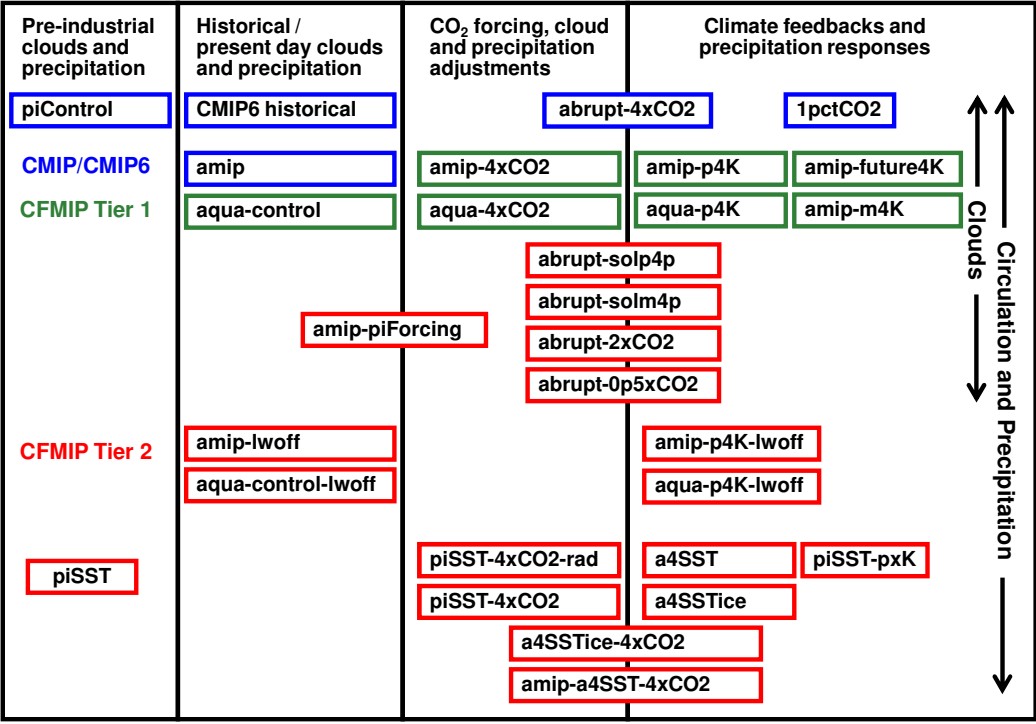

**Figure 1.** Summary of CFMIP-3 experiments and CMIP DECK / CMIP6 experiments.



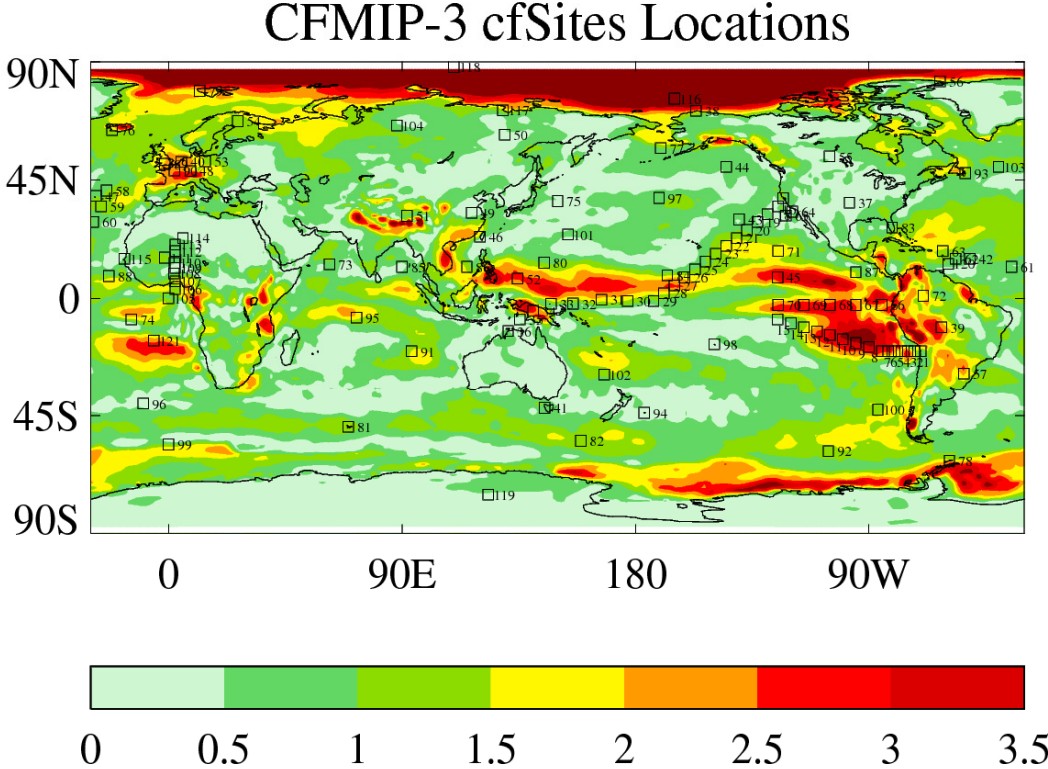

**Figure 2**. CFMIP-3 cfSites locations. The contours give an indication of inter-model spread in cloud feedback from the
CFMIP-2 amip/amip4K experiments (please refer to Webb et al., 2015a for details).





**Table 1.** Summary of CFMIP Tier 1 experiments.

| Experiment Name | Experiment Description / Design | Configuration | Start Year | Length |
|---|---|---|---|---|
| amip | This a single ensemble member of the AMIP DECK experiment which contains additional outputs which are required for model evaluation using COSP, and as control values for model outputs in the amip-p4K, amip-4xCO2, amip-future4K and amip-m4K experiments. | Atmos-only | 1979 | 36 |
| amip-p4K | As CMIP5/CFMIP-2 amip4K experiment. AMIP experiment where SSTs are subject to a uniform warming of 4K. | Atmos-only | 1979 | 36 |
| amip-4xCO2 | As CMIP5/CFMIP-2 amip4xCO2 experiment. AMIP experiment where SSTs are held at control values and the $CO_2$ seen by the radiation scheme is quadrupled. | Atmos-only | 1979 | 36 |
| amip-future4K | As CMIP5/CFMIP-2 amipFuture experiment. AMIP experiment where SSTs are subject to a composite SST warming pattern derived from coupled models, scaled to an ice-free ocean mean of 4K. | Atmos-only | 1979 | 36 |
| aqua-control | Extended version of CMIP5/CFMIP-2 aquaControl experiment. Aquaplanet (no land) experiment with no seasonal cycle forced with specified zonally symmetric SSTs. | Atmos-only | 1979 | 10 |
| aqua-p4K | Extended version of CMIP5/CFMIP-2 aqua4K experiment. Aquaplanet experiment where SSTs are subject to a uniform warming of 4K. | Atmos-only | 1979 | 10 |
| aqua-4xCO2 | Extended version of CMIP5/CFMIP-2 aqua4xCO2 experiment. Aquaplanet experiment where SSTs are held at control values and the $CO_2$ seen by the radiation scheme is quadrupled. | Atmos-only | 1979 | 10 |






**Table 2.** Summary of CFMIP Tier 2 experiments.

| Experiment Name | Experiment Description / Design | Configuration | Start Year | Length |
|---|---|---|---|---|
| amip-m4K | As amip experiment but SSTs are subject to a uniform cooling of 4K. | Atmos-only | 1979 | 36 |
| amip-lwoff | As amip experiment, but with cloud-radiative effects switched off in the LW radiation code. | Atmos-only | 1979 | 36 |
| amip-p4K-lwoff | As amip-p4K experiment, but with cloud-radiative effects switched off in the LW radiation code. | Atmos-only | 1979 | 36 |
| aqua-control-lwoff | As aqua-control experiment, but with cloud-radiative effects switched off in the LW radiation code. | Atmos-only | 1979 | 10 |
| aqua-p4K-lwoff | As aqua-p4K experiment, but with cloud-radiative effects switched off in the LW radiation code. | Atmos-only | 1979 | 10 |
| abrupt-solp4p | Conceptually similar to abrupt 4xCO2 DECK experiment, except that the solar constant rather than $CO_2$ is abruptly increased by 4%. | Coupled AOGCM | 1850 | 150 |
| abrupt-solm4p | Same as abrupt-solp4p, except solar constant is reduced by 4% rather than increased. | Coupled AOGCM | 1850 | 150 |
| abrupt-2xCO2 | Identical to the DECK abrupt4xCO2, but at $2xCO_2$. | Coupled AOGCM | 1850 | 150 |
| abrupt-0p5xCO2 | Identical to the DECK abrupt4xCO2, but at $0.5xCO_2$ | Coupled AOGCM | 1850 | 150 |
| amip-piForcing | Identical to AMIP DECK experiment but from 1870-present with constant pre-industrial forcing levels (anthro & natural). | Atmos-only | 1870 | 145 |
| piSST | An AGCM experiment with monthly-varying SSTs, sea-ice, atmospheric constituents and any other necessary boundary conditions (e.g. vegetation if required) taken from each model's own piControl run (using the 30 years of piControl that are parallel to years 111-140 of its abrupt4xCO2 run). Dynamic vegetation should be turned off in all the piSST set of experiments. | Atmos-only | Year 111 of abrupt-4xCO2 | 30 |
| piSST-pxK | Same as piSST, but with a spatially and temporally uniform SST anomaly applied on top of the monthly-varying piSST SSTs. The magnitude of the uniform increase is taken from each model's global, climatological annual mean SST change between abrupt4xCO2 minus piControl (using the mean of years 111-140 of abrupt4xCO2, and the parallel 30-year section of piSST). | Atmos-only | Year 111 of abrupt-4xCO2 | 30 |
| piSST-4xCO2-rad | Same as piSST but $CO_2$ as seen by the radiation scheme is quadrupled. | Atmos-only | Year 111 of abrupt-4xCO2 | 30 |
| piSST-4xCO2 | Same as piSST but $CO_2$ is quadrupled. The increase in $CO_2$ is seen by both the radiation scheme and vegetation. | Atmos-only | Year 111 of abrupt-4xCO2 | 30 |
| a4SST | As piSST, but with monthly-varying SSTs taken from years 111-140 of each model's own abrupt4xCO2 experiment instead of from piControl. Sea-ice is unchanged from piSST. | Atmos-only | Year 111 of abrupt-4xCO2 | 30 |
| a4SSTice | As piSST, but with monthly-varying SSTs and sea-ice taken from years 111-140 of each model's own abrupt4xCO2 experiment instead of from piControl. | Atmos-only | Year 111 of abrupt-4xCO2 | 30 |
| a4SSTice-4xCO2 | As a4SSTice, but CO2 is quadrupled, and the increase in CO2 is seen by both the radiation scheme and vegetation. | Atmos-only | Year 111 of abrupt-4xCO2 | 30 |
| amip-a4SST-4xCO2 | Same as amip, but a patterned SST anomaly is applied on top of the monthly-varying amip SSTs. This anomaly is a monthly climatology, taken from each model's own abrupt4xCO2 run minus piControl (using the mean of years 111-140 of abrupt4xCO2, and the parallel 30-year section of piControl). $CO_2$ is quadrupled, and the increase in $CO_2$ is seen by both the radiation scheme and vegetation. | Atmos-only | 1979 | 36 |







**Table 3.** Summary of CFMIP-OBS observational datasets available for comparison with COSP diagnostics.

| Dataset | Years | Observables | Applications | References |
|---|---|---|---|---|
| CALIPSO-GOCCP | 2006/06 - 2012/10 | Cloud fractions: 2D and 3D by phase. Scattering ratio histograms as function of height. | Vertical distributions of clouds. Cloud phase identification. | Chepfer et al., (2010); Cesana and Chepfer, (2013) |
| CloudSat | 2006/06 - 2010/12 | Reflectivity histograms as function of height. | Vertical distributions of clouds and precipitation | Marchand et al., (2009); Zhang et al., (2010) |
| ISCCP | 1983/07-2008/06 | Cloud top pressure – cloud optical depth histograms. | Cloud radiative properties. Long time series. | Rossow and Schiffer, (1999) |
| MODIS | 2002/07 – 2015/11 | Cloud top pressure – cloud optical depth histograms. Total, liquid and ice cloud fractions. Effective radius – optical depth histograms by cloud phase. | Cloud radiative properties. Effective size, and phase information. | Pincus et al., (2012); King et al., (2003) |
| MISR | 2000/06 – 2013/05 | Cloud top height (CTH) – cloud optical depth histograms | Cloud radiative properties. Independent estimate of cloud top height. | Marchand et al., (2010) |
| PARASOL | 2003/05 - 2012/08 | Monodirectional reflectance | Cloud radiative properties. | Konsta et al., (2015) |

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
