# Peer review of "The Cloud Feedback Model Intercomparison Project (CFMIP) contribution to CMIP6."

_Geoscientific Model Development, 2016_

## Short Comment (SC1) · 18 May 2016

This paper provides a clear description of the design of CFMIP3/CMIP6. The proposed experiments and outputs are interesting and will be important contributions to CMIP6. I have only a few minor comments.

I assume that all the CFMIP experiments are $CO_2$ concentration driven. Should ESMs turn off dynamic vegetation and chemistry schemes?

Line 213 "Sea ice and SSTs under sea ice remain the same as in the amip DECK experiment.": How should we set SSTs in grids with 50% concentration of sea ice?

Line 263 "As such we hope that these experiments will provide useful synergies with Palaeoclimate Model Intercomparision Project (PMIP)": If there are any experiments

that are directly related to the CFMIP experiments, please specify.

Line 302 "cloud-radiative effects are switched off in the longwave part of the radiation code": Is the shortwave part retained?

2.4 Abrupt +/-4% solar forced runs: Not only TSI but also spectral solar irradiance (SSI) are provided for CMIP6 (http://solarisheppa.geomar.de/cmip6). I assume that many ESMs use the SSI data for their DECK experiments. How to add +/-4% solar forcing on SSI?

Line 411 piSST: Do we use the monthly mean values of each year of piControl? Monthly mean climatology would lead to better S/N.

Line 550 "allowing a detailed evaluation clouds": allowing a detailed evaluation of clouds?

Hope this helps.

---

## Short Comment (SC2) · 25 May 2016

This paper summarizes the objectives of CFMIP and the contribution of CFMIP-3 to CMIP6. CFMIP helps to explain the spread of cloud feedbacks, adjustments and processes across climate models. This updated contribution goes a step forward and suggests additional experiments to allow the community to tackle in more detail the physical reasons underlying dynamical and regional biases seen in climate models. By proposing experiments that test especially the atmospheric components of climate models, CFMIP provides a relevant framework to understand and improve cloud parameterizations and processes which remain the principal sources of surface and atmospheric model biases.

First, the authors summarised well how former CFMIP/CMIP5 experiments helped to

improve our scientific understanding of climate feedbacks. It thus provides a relevant background supporting the additional experiments that they advise the modelling groups to perform. I particularly appreciated (1) the will to promote the analysis of experiments when cloud radiative effects are switched off, (2) the pertinent time slice experiments aiming to understand regional climate responses and (3) the encouragement of a more extensive distribution and use of physical tendencies which are a signature of the atmospheric components of climate models.

Below, I have listed a number of minor points which might be addressed to clarify the text (if the authors find them useful):

- Some acronyms are not defined : AOGCM (l.83), GCM (l.92), RFMIP (l.378), TOA (l.637), PMC (l.777)

L. 196 : I have trouble understanding the meaning of "known answer".

L. 217: The amip-future4K experiments used the CMIP3 pattern of SST increase. Is this pattern consistent with the one derived from CMIP5 models?

L. 222-225 and L. 419-422: I'm a little bit confused about all 4xCO2 experiments. The amip4xCO2 experiment involves the CO2 effect on the atmospheric component and land warming without the vegetation feedback. It is thus "equivalent" to the piSST-4xCO2-rad experiment listed in section 2.7 (but not to piSST-4xCO2). I guess abrupt4xCO2 takes into account the vegetation feedback. So, the amip4xCO2 experiment should be named amip4xCO2-rad, doesn't it?

L.257-264: You could also add the reference "Block and Mauristen (13) JAMES - Forcing and Feedback in the MPI-ESM-LR coupled model under abruptly quadrupled CO2", which highlights the utility of diverse amip-pXk and abrupt2xCO2 experiments.

L. 288-299:

(1) It is thus right that LW effects are the most important contributor to cloud atmospheric radiative effects, and SW effects play a minor role (e.g. Takahashi 09). Nevertheless, local SW cloud effects exist (Pendergrass and Hartmann, 14). It might thus be interesting to point this fact out in the text and leave the discussion about SW effects sufficiently open.

(2) Since only LW radiative effects are removed, does it mean that models still have a SW cloud feedback but no LW cloud feedback?

(3) "and the radiation code only". Does this mean that, for instance, a boundary-layer parameterization based on LW cloud-top radiative cooling continues to see LW effects?

L.326-328: Contrary to CO2 effects, the radiative forcing of solar insolation depends on latitude. Is this dependency taken into account when the authors state that a 4% change results in a "radiative forcing of a similar magnitude to that due to CO2 quadrupling"?

L. 482: Single Column Model already defined line 91-92.

L. 600-601: Is it normal that "cfDay-2d" is named by CMIP5 and not CFMIP? Why is there no CMIP5 or CFMIP prefix for "cfDay-3d"?

Fig.1 : The DECK is written in the caption but not highlighted in the graph.

Fig.1: I consider lwoff experiments as part of the "Clouds" analysis. You may consider making the arrow longer.

——————————

---

## Referee Comment (RC1) · Anonymous Referee #1 · 15 Jun 2016

In this paper, the authors state the goals and motivation of CFMIP, review the major accomplishments of previous CFMIPs, and describe the proposed experiments and diagnostics for CFMIP3. The coordinated experiments proposed for CFMIP3 will target a number of outstanding questions for which previous model intercomparisons were not equipped to address, in addition to sustaining a number of highly useful experiments from earlier MIPs that will help to characterize and understand the response of the CMIP6-generation of models to external forcing (in addition to help quantify the forcing itself). Advanced diagnostics (e.g., satellite simulators and high frequency tendency terms) will aid in dissecting model results, and the authors have proposed that they be used more broadly (e.g., COSP turned on for longer durations and in more experiments). The emphasis on (mostly) atmosphere-only simulations in CFMIP3 should hopefully make it appealing for modeling centers to take part in several of the experiments despite the high volume of requested diagnostics.

The scientific questions to be addressed by CFMIP3 are well articulated and the various proposed experiments seem well designed to address these questions, and will advance the community's knowledge. The presentation of the paper is not particularly concise, not are the figures particularly insightful, but the writing is clear and overall the presentation seems appropriate for a paper proposing a model intercomparison project. Thus, in my opinion the manuscript represents a substantial contribution to modelling science within the scope of Geoscientific Model Development, and I recommend publication following consideration of some minor comments detailed below.

Specific Comments:

*piSST and a4SST: It is not clear to me whether (a) monthly- and annually-varying SSTs from the relevant 30 years in the piControl run, or (b) a monthly-resolved climatology of SSTs over the relevant 30 years in the piControl run are prescribed in piSST. Same question for a4SST.

*amip-piForcing: I'm curious whether there was any interest in performing a similar experiment, but with present-day (rather than preindustrial) forcing held fixed. An example application that occurs to me is that a model with large aerosol-cloud interactions would presumably have brighter clouds with smaller droplets downwind of aerosol sources if the forcing were fixed at present-day, and its temperature-mediated changes in clouds might therefore be different than that occurring in an atmosphere with fewer aerosols. Having these two experiments would allow one to explore this effect (and others related to other forcing agents).

*Given its implications for understanding apparent state- or time-dependent changes in effective climate sensitivity, I was a little surprised to see no experiments designed to explore causes of nonlinearity in the Gregory plot, perhaps using warming experiments in which the SST pattern is fixed in time (with various patterns), similar to those conducted in Andrews et al, J. Climate (2015). Is there a reason for not proposing these,

or are these effects already captured in other proposed experiments?

*Line 126: should be "...meetings AND international..."

*Line 426: "a4SST-4xCO2-all" should be "a4SSTice-4xCO2-all". There may be other instances of this; please verify that they are also changed.

*Line 512: What is the reason for dispensing with the cloud tendency terms in CFMIP3?

*Lines 597-608: it is not clear to me why some of these have a CMIP5 prefix, a CFMIP prefix, or no prefix at all (cfDay-3d). Why would a CMIP5 prefix be appropriate at all?

*Line 611: should be "...for 140 years OF the piControl..."

*Appendix A: I don't understand what is meant by "Lead coordinator". Is this the person who has "first dibs" on writing papers based on these experiments? Are interested investigators expected to contact this person to avoid duplicating work that others are doing with output from these experiments?

*Figure 1: I think "CMIP6" should be deleted before "historical". If it is supposed to be there, I don't understand why it is only there. It is also unclear to me why the "Clouds" arrow only extends as far as abrupt-0p5xCO2. I think both the clouds arrow and the circulation and precipitation arrows should include all experiments, but in that case, what is the point of showing them?

*Table 1: should be "This IS a single..."

*Table 3: Several of the observational datasets end many years ago despite the fact that these satellites are still in orbit. Are there plans to extend these records, especially since the AMIP runs end in 2015?

---

## Referee Comment (RC2) · Anonymous Referee #2 · 21 Jun 2016

This manuscript outlines the CFMIP-3 experimental strategies, the associated model output, and the motivation and anticipated results of these experiments. Overall the manuscript is clearly written and accurately summarizes the plans for CFMIP-3, and in many ways represents more of a review of past CFMIP achievements, which in itself is a useful contribution. I recommend acceptance with only minor revisions as outlined in my suggestions below.

The authors are very generous in their citations of other work, which is commendable, but it detracts from the readability of the manuscript. I recommend the authors consider focusing on a few select highlights of the previous CFMIPs that illustrate the main contributions, rather than attempting an exhaustive summary of everything that's been learned from CFMIP experiments. In the current form, it's difficult to identify what the key contributions of CFMIP have been.

[Figure]

Section 2.1 reads more as a review of all previous studies that used CFMIP data, rather than a description of the CFMIP-3. I would recommend moving much of this to the previous section which reviews past CFMIPs and identify any changes/deletions from the past CFMIPs before then proceeding to describe the new additions to the CFMIP-3 set of experiments. It would also be useful to define what a "DECK" is.

There is rightfully considerable attention within CFMIP devoted to isolating and quantifying the fast adjustments. However the fast adjustments arise from both atmospheric radiative heating changes and land warming. It would be useful to isolate these contributions (beyond the use of aqua planets, whose utility in quantifying CGCM feedbacks is a little over sold here IMO). Has there been any efforts to develop experiments for this? If not, this issue might warrant some discussion in reference to the experiments designed to quantify adjustments.
* * *

---

## Referee Comment (RC3) · A. Voigt (Referee) · 27 Jul 2016

The authors provide a concise and well-written presentation of the CFMIP experiments proposed for CMIP6, which will continue the successful CFMIP activities over the last 15 years. I enjoyed reading the paper, in particular the historical context given in the introduction, and find that it nicely presents the scientific motivation and chosen simulation strategy at a level amenable to both CFMIP experts and climate scientists with other backgrounds. I recommend publication in GMD after my following minor comments have been addressed.

Line 217, amip-future4K simulations: Why is the CMIP3 SST pattern used and not an updated pattern from CMIP5 AOGCM runs?

Line 227: I am very glad to hear that the CMIP-3 aquaplanet simulations will be extended to 10 years. This will be beneficial for studies of extratropical dynamics, for which internal variability is larger than in the tropics.

Line 238, amip-m4k simulations: I am wondering to what extent some models might have problems with SSTs below freezing? Maybe this might require code changes in some models in case they employ a fixed lower threshold for the SST used in the calculation of surface fluxes? Such a problem would, of course, not occur for the p4K simulations?

Lines 279: The authors might consider to also refer to Voigt and Shaw (2015, Nature Geoscience) here for the extratropical circulation. The study showed that cloud-radiative feedbacks contribute substantially to the poleward jet shifts under 4K warming in aquaplanet simulations.

Line 266, lwoff experiments: Just an idea, but I though it's worthwhile bringing it up here: While the surface cloud effect is stronger in the shortwave than the longwave domain, the longwave can still substantial. I am wondering whether an experiment with clear-sky heating in the atmosphere and all-sky heating at the surface would be even better to isolate the effect of atmospheric cloud-radiative heating. I suspect it's too late to change the experimental protocol, and maybe there is a reason why lwoff is still better. If so, it might be worthwhile to briefly discuss this.

Line 342: Non-linearity was also shown in the CMIP5 ensemble by Meraner et al. (2013, GRL, doi:10.1002/2013GL058118). Meraner et al. showed non-linear climate sensitivity across the multi-model CMIP5 ensemble, whereas the other cited work used single models if I am not mistaken. So maybe worthwhile including here?

Line 368: Maybe specify the reason why the CFMIP2/CMIP5 runs did not allow such an estimate. I.e., I assume that one would use SST-driven simulations for this and that the usual amip period is too short to reliably calculate feedbacks?

Sect. 2.7: The time slice experiments ask an interesting question but given that 8

experiments are demanded, I was wondering how they ought to be combined to answer the questions in mind. Maybe the authors can give an example?

Line 487: I would be curious to know about the reasons to no longer ask for cfSites output in the aquaplanet ensemble and amip-future4K. Is it the lack of observational data to compare to, or a choice to avoid asking for too much data?

Line 685: and –> an

Figure 1: Why does the vertical cloud bar on the right side not include the lwoff simulations?

For some of the proposed simulations the link to clouds, which are the prime motivation for CFMIP, is not very evident and maybe could be made clearer? I am thinking of the simulations in Sect. 2.7 (time slice experiments) and Sect. 2.5 (nonLinMIP).

---

## Editor Comment (EC1) · J. C. Hargreaves (Editor) · 4 Aug 2016

This seems to me to be a pretty good MIP manuscript. You do have the advantage that the protocols for the experiments are relatively simple to describe, but I still think it very well organised. I haven't checked all the protocols - just a couple that I am particularly interested in - but the information seemed complete for those. However, please do check through your revised manuscript to make sure that a third party could set up each run from the information provided. The remaining peculiarity is the reference to boundary conditions that will become available through other papers in this special issue - are there now references that can be provided for these papers?

The thing I spotted in the reviewers' comments that I am unsure about is the suggestion to abbreviate the citations to increase readability. Here's an example,

"Temperature and humidity tendency terms in particular have been shown to be useful for understanding the roles of different parts of the model physics in cloud feedbacks and adjustments (Kamae and Watanabe 2012; Williams et al., 2013; Webb and 509 Lock 2013; Demoto et al., 2013; Sherwood et al., 2014; Ogura et al., 2014; Brient et al., 2015) "

I generally don't like the idea of reducing the citations, but that is an awful lot of references all apparently showing the same thing! As a reader I'd want to know what the difference is between these papers, and which one I should look up in order to learn about the thing I am specifically interested in. The obvious solution would be to add a little more description, so that the reader has more knowledge about the content of the references. Doing so will make the manuscript longer, which could get out of hand, but maybe there is a middle way which produces a more readable and more useful manuscript...?

———————————————

---

## Author Comment (AC1) · 15 Sep 2016

Reviewer comments below are shown in bold and our responses are in italics.

*Dear Hideo,*

**This paper provides a clear description of the design of CFMIP3/CMIP6. The proposed experiments and outputs are interesting and will be important contributions to CMIP6. I have only a few minor comments.**

*Thank you for your careful consideration of our manuscript and for these helpful comments.*

**I assume that all the CFMIP experiments are CO2 concentration driven. Should ESMs turn off dynamic vegetation and chemistry schemes?**

*The CFMIP experiments are indeed driven by CO2 concentration rather than CO2 emissions. Many of the CFMIP experiments are based on the DECK experiments (e.g. amip, piControl, abrupt-4xCO2). Experiments such as amip-p4K and abrupt-2xCO2 should be configured consistently with the DECK experiments that they are based on.*

*We will add additional text in the 2$^{nd}$ line of Section 2 as follows:*

*"Most of the CFMIP-3 experiments are based on $CO_2$ concentration forced amip, piControl and abrupt-4xCO2 CMIP DECK (Diagnostic, Evaluation and Characterization of Klima) experiments (Eyring et al., 2016). Unless otherwise specified below, the CFMIP-3 experiments should be configured consistently with the DECK experiments on which they are based, using consistent model formulation, and forcings and boundary conditions as specified by Eyring et al., 2016."*

**Line 213 "Sea ice and SSTs under sea ice remain the same as in the amip DECK experiment.": How should we set SSTs in grids with 50% concentration of sea ice?**

*We will modify this text as follows:*

*"Sea ice and SSTs in grid boxes containing sea ice remain the same as in the amip DECK experiment."*

**Line 263 "As such we hope that these experiments will provide useful synergies with Palaeoclimate Model Intercomparision Project (PMIP)": If there are any experiments that are directly related to the CFMIP experiments, please specify.**

*We will modify this text as follows:*

*"As such we hope that these experiments will provide useful synergies with the Palaeoclimate Model Intercomparision Project (PMIP) CMIP6 experiments (e.g. in interpreting differing cloud feedbacks*

*between future $CO_2$ forced experiments and those representing the Last Glacial Maximum, as highlighted by Yoshimori et al., 2009)."*

**Line 302 "cloud-radiative effects are switched off in the longwave part of the radiation code": Is the shortwave part retained?**

*We will modify this text as follows:*

*"cloud-radiative effects are switched off in the longwave part of the radiation code while retaining those in the shortwave."*

**2.4 Abrupt +/-4% solar forced runs: Not only TSI but also spectral solar irradiance (SSI) are provided for CMIP6 (http://solarisheppa.geomar.de/cmip6). I assume that many ESMs use the SSI data for their DECK experiments. How to add +/-4% solar forcing on SSI?**

*We will add a line to section 2.4 which states:*

*" When changing the solar constant, the shape of the spectral solar irradiance distribution should remain consistent with that in the piControl experiment."*

**Line 411 piSST: Do we use the monthly mean values of each year of piControl? Monthly mean climatology would lead to better S/N.**

*We will add the following to Section 2.7*

*"These are forced with monthly- and annually-varying monthly mean SSTs and sea ice, which reproduce regional precipitation patterns more accurately than is possible using climatological SST forcing."*

**Line 550 "allowing a detailed evaluation clouds": allowing a detailed evaluation of clouds?**

*We will correct that.*

---

## Author Comment (AC2) · 15 Sep 2016

Reviewer comments below are shown in bold and our responses are in italics.

*Dear Florent,*

**This paper summarizes the objectives of CFMIP and the contribution of CFMIP-3 to CMIP6. CFMIP helps to explain the spread of cloud feedbacks, adjustments and processes across climate models. This updated contribution goes a step forward and suggests additional experiments to allow the community to tackle in more detail the physical reasons underlying dynamical and regional biases seen in climate models.**
**By proposing experiments that test especially the atmospheric components of climate models, CFMIP provides a relevant framework to understand and improve cloud parameterizations and processes which remain the principal sources of surface and atmospheric model biases.**
**First, the authors summarised well how former CFMIP/CMIP5 experiments helped to improve our scientific understanding of climate feedbacks. It thus provides a relevant background supporting the additional experiments that they advise the modelling groups to perform. I particularly appreciated (1) the will to promote the analysis of experiments when cloud radiative effects are switched off, (2) the pertinent time slice experiments aiming to understand regional climate responses and (3) the encouragement of a more extensive distribution and use of physical tendencies which are a signature of the atmospheric components of climate models.**

*Thank you for your careful consideration of our manuscript and for these helpful comments.*

**Below, I have listed a number of minor points which might be addressed to clarify the text (if the authors find them useful)**

**- Some acronyms are not defined : AOGCM (l.83), GCM (l.92), RFMIP (l.378), TOA (l.637), PMC (l.777)**

*We will define these in the revised manuscript.*

**L. 196 : I have trouble understanding the meaning of "known answer".**

*We will modify this sentence to read:*

*"Aqua-planet simulations (and other idealized) experiments are particularly effective at highlighting model differences, for instance in the placement of the tropical rain bands, or in the representation of cloud changes with warming, as it is not possible to tune them to observations in the same way as is for more realistic configurations (e.g., Stevens and Bony, 2013)."*

**L. 217: The amip-future4K experiments used the CMIP3 pattern of SST increase. Is this pattern consistent with the one derived from CMIP5 models?**

*We haven't looked into this, because we consider consistency with the CMIP5 protocol to be more important than using SSTs from CMIP5 rather than CMIP3.*

*We will add the following to Appendix C:*

*"We have retained the SST forcing based on the CMIP3 coupled models because we consider it more important to be able to compare CMIP5 and CMIP6 models forced with the same SST pattern than to use a pattern which is consistent with, say, the CMIP5 coupled response."*

**L. 222-225 and L. 419-422: I'm a little bit confused about all 4xCO2 experiments. The amip4xCO2 experiment involves the CO2 effect on the atmospheric component and land warming without the vegetation feedback. It is thus "equivalent" to the piSST-4xCO2-rad experiment listed in section 2.7 (but not to piSST-4xCO2). I guess abrupt4xCO2 takes into account the vegetation feedback. So, the amip4xCO2 experiment should be named amip4xCO2-rad, doesn't it?**

*We agree that this would be a more consistent naming of this experiment. However, we think that the experiment descriptions are clear. Unfortunately however we understand that CMIP6 experiment names have now been finalised and propagated to the ESG and so it is not now possible to change them.*

**L.257-264: You could also add the reference "Block and Mauristen (13) JAMES - Forcing and Feedback in the MPI-ESM-LR coupled model under abruptly quadrupled CO2", which highlights the utility of diverse amip-pXk and abrupt2xCO2 experiments. L. 288-299:**

*We will add a citation to this paper in section 2.5.*

**(1) It is thus right that LW effects are the most important contributor to cloud atmospheric radiative effects, and SW effects play a minor role (e.g. Takahashi 09). Nevertheless, local SW cloud effects exist (Pendergrass and Hartmann, 14). It might thus be interesting to point this fact out in the text and leave the discussion about SW effects sufficiently open.**

*We will add the following to section 2.3:*

*"We note that the presence of clouds does affect the shortwave radiative heating of the atmosphere, although this is a much smaller effect than its longwave equivalent (e.g. Pendergrass and Hartmann, 2014)."*

**(2) Since only LW radiative effects are removed, does it mean that models still have a SW cloud feedback but no LW cloud feedback?**

*Yes. We will clarify this by adding the following to section 2.3:*

*"In this configuration, the models will have a shortwave cloud feedback but no longwave cloud feedback."*

**(3) "and the radiation code only". Does this mean that, for instance, a boundary-layer parameterization based on LW cloud-top radiative cooling continues to see LW effects?**

*We will add the following comment to section 2.3:*

*"Care should also be taken to remove the effects of cloud on any longwave cooling used in other model schemes (e.g. turbulent mixing) if these are calculated independently of the radiation scheme. "*

**L.326-328: Contrary to CO2 effects, the radiative forcing of solar insolation depends on latitude. Is this dependency taken into account when the authors state that a 4% change results in a "radiative forcing of a similar magnitude to that due to CO2 quadrupling"?**

*Yes this has been taken into account. We will modify the text as follows to make it clear that this gives a similar magnitude in global mean forcing.*

*"...resulting in a global mean radiative forcing of a similar magnitude to that due to $CO_2$ quadrupling."*

**L. 482: Single Column Model already defined line 91-92.**

*Duplication removed.*

**L. 600-601: Is it normal that "cfDay-2d" is named by CMIP5 and not CFMIP? Why is there no CMIP5 or CFMIP prefix for "cfDay-3d"?**

*The different prefixes represent detail in the formal data request which is not required here. In the manuscript we will delete the prefixes to avoid confusion, and will add the following sentence:*

*"(Please note that in the full data request these variable groups are in many cases split into a number of sub-tables. As noted above, the formal data request provides the definitive specification of the model outputs.)"*

**Fig.1 : The DECK is written in the caption but not highlighted in the graph.**

*We will update the figure and caption to be consistent in this regard.*

**Fig.1: I consider lwoff experiments as part of the "Clouds" analysis. You may consider**

**making the arrow longer.**

*We will do this.*

---

## Author Comment (AC6) · 15 Sep 2016

**Response to Interactive comment by J. C. Hargreaves (Editor) on "The Cloud Feedback Model Intercomparison Project (CFMIP) contribution to CMIP6" by Mark J. Webb et al.**

Reviewer comments below are shown in bold and our responses are in italics.

*Dear Julia,*

**This seems to me to be a pretty good MIP manuscript. You do have the advantage that the protocols for the experiments are relatively simple to describe, but I still think it very well organised.**

*Thank you.*

**I haven't checked all the protocols - just a couple that I am particularly interested in - but the information seemed complete for those. However, please do check through your revised manuscript to make sure that a third party could set up each run from the information provided.**

*We have done this, and have made a few changes to clarify some issues:*

*We have updated the amip-future4K experiment definition in Section 2.1 to include the sentence:*

*"Care should be taken to ensure that SSTs are increased in any inland bodies of water and near coastal edges, for example by linearly interpolating the provided warming pattern dataset to fill in missing data before re-gridding to the target resolution. "*

*In section 2.7 we have inserted the word 'open' into the sentence:*

*"The magnitude of the uniform increase is taken from each model's global, climatological annual mean open SST change between abrupt-4xCO2 and piControl (using the mean of years 111-140 of abrupt-4xCO2, and the parallel 30-year section of piControl)."*

*We have also re-written part of Section 2.6 to make it clearer:*

*"Time-varying feedbacks in the amip experiment could alternatively be diagnosed by subtracting a time-varying radiative forcing diagnosed from RFMIP experiments. However, the amip-piForcing approach has the benefit of diagnosing the time-varying feedbacks over the full 1870-present period rather than the last 36 years, and does so with reference to a single experiment, which reduces noise compared to that which would be present with a double difference of the amip experiment and two RFMIP experiments."*

*We have also added the following text to Appendix B:*

*"Ozone values are provided up to 0.28hPa (about 60km altitude in mid-latitudes). For models with tops above this level, we recommend that the value at the top level in the forcing dataset is applied*

*above 0.28hPa so as to give a profile which is constant with height above 0.28hPa. We are currently looking into providing an alternative ozone forcing dataset with a higher top. If such a dataset becomes available we will publish it via the CFMIP website."*

**The remaining peculiarity is the reference to boundary conditions that will become available through other papers in this special issue - are there now references that can be provided for these papers?**

*We will add the following:*

*"Most of the CFMIP-3 experiments are based on $CO_2$ concentration forced amip, piControl and abrupt-4xCO2 CMIP DECK (Diagnostic, Evaluation and Characterization of Klima) experiments (Eyring et al., 2016). Unless otherwise specified below, the CFMIP-3 experiments should be configured consistently with the DECK experiments on which they are based, using consistent model formulation, and forcings and boundary conditions as specified by Eyring et al., 2016."*

**The thing I spotted in the reviewers' comments that I am unsure about is the suggestion to abbreviate the citations to increase readability. Here's an example,**
**"Temperature and humidity tendency terms in particular have been shown to be useful for understanding the roles of different parts of the model physics in cloud feedbacks and adjustments (Kamae and Watanabe 2012; Williams et al., 2013; Webb and 509 Lock 2013; Demoto et al., 2013; Sherwood et al., 2014; Ogura et al., 2014; Brient et al., 2015) "**
**I generally don't like the idea of reducing the citations, but that is an awful lot of references all apparently showing the same thing! As a reader I'd want to know what the difference is between these papers, and which one I should look up in order to learn about the thing I am specifically interested in. The obvious solution would be to add a little more description, so that the reader has more knowledge about the content of the references. Doing so will make the manuscript longer, which could get out of hand, but maybe there is a middle way which produces a more readable and more useful manuscript.**

*We have addressed this in the manuscript as follows. We have reduced the number of citations in the introduction, in particular where there is duplication with Section 2. Throughout, where several citations are made together, we have broken them into smaller groups as suggested to give the reader a better idea of what distinguishes them.*

*For example, in the case highlighted above, we have updated the manuscript to read:*

*"Temperature and humidity tendency terms in particular have been shown to be useful for understanding the roles of different parts of the model physics in cloud feedbacks (e.g. Webb and Lock 2013; Demoto et al., 2013; Sherwood et al., 2014; Brient et al., 2015) and cloud adjustments (e.g. Kamae and Watanabe 2012; Ogura et al., 2014) as well as in understanding clouds and circulation in the present climate (e.g. Williams et al., 2013; Oueslati and Bellon, 2013; Xavier et al., 2015). "*

*We hope that these changes strike the required balance effectively.*

---

## Author Response (AR1)

Reviewer comments below are shown in bold and our responses are in italics.

*Dear Hideo,*

**This paper provides a clear description of the design of CFMIP3/CMIP6. The proposed experiments and outputs are interesting and will be important contributions to CMIP6. I have only a few minor comments.**

*Thank you for your careful consideration of our manuscript and for these helpful comments.*

**I assume that all the CFMIP experiments are CO2 concentration driven. Should ESMs turn off dynamic vegetation and chemistry schemes?**

*The CFMIP experiments are indeed driven by CO2 concentration rather than CO2 emissions. Many of the CFMIP experiments are based on the DECK experiments (e.g. amip, piControl, abrupt-4xCO2). Experiments such as amip-p4K and abrupt-2xCO2 should be configured consistently with the DECK experiments that they are based on.*

*We have added additional text at line 176 as follows:*

*"Most of the CFMIP-3 experiments are based on CO$_2$ concentration forced amip, piControl and abrupt-4xCO2 CMIP DECK (Diagnostic, Evaluation and Characterization of Klima) experiments (Eyring et al., 2016). Unless otherwise specified below, the CFMIP-3 experiments should be configured consistently with the DECK experiments on which they are based, using consistent model formulation, and forcings and boundary conditions as specified by Eyring et al., 2016."*

**Line 213 "Sea ice and SSTs under sea ice remain the same as in the amip DECK experiment.": How should we set SSTs in grids with 50% concentration of sea ice?**

*We have modified the text at L268 as follows.  L269, L356 also amended similarly.*

*"Sea ice and SSTs in grid boxes containing sea ice remain the same as in the amip DECK experiment."*

**Line 263 "As such we hope that these experiments will provide useful synergies with Palaeoclimate Model Intercomparision Project (PMIP)": If there are any experiments that are directly related to the CFMIP experiments, please specify.**

*We have modified this text as follows (L361):*

*"As such we hope that these experiments will provide useful synergies with the Palaeoclimate Model Intercomparision Project (PMIP) CMIP6 experiments (e.g. in interpreting differing cloud feedbacks between future $CO_2$ forced experiments and those representing the Last Glacial Maximum, as highlighted by Yoshimori et al., 2009)."*

**Line 302 "cloud-radiative effects are switched off in the longwave part of the radiation code": Is the shortwave part retained?**

*We have modified this text as follows: (L409)*

"cloud-radiative effects are switched off in the longwave part of the radiation code while retaining those in the shortwave (Fermepin and Bony, 2014)."

**2.4 Abrupt +/-4% solar forced runs: Not only TSI but also spectral solar irradiance (SSI) are provided for CMIP6 (http://solarisheppa.geomar.de/cmip6). I assume that many ESMs use the SSI data for their DECK experiments. How to add +/-4% solar forcing on SSI?**

*We have added a line to section 2.4 which states: (L439)*

*" When changing the solar constant, the shape of the spectral solar irradiance distribution should remain consistent with that in the piControl experiment."*

**Line 411 piSST: Do we use the monthly mean values of each year of piControl? Monthly mean climatology would lead to better S/N.**

*We have added the following to Section 2.7 (L550)*

*"These are forced with monthly- and annually-varying monthly mean SSTs and sea ice, which reproduce regional precipitation patterns more accurately than is possible using climatological SST forcing (Skinner et al., 2012)."*

**Line 550 "allowing a detailed evaluation clouds": allowing a detailed evaluation of clouds?**

*We have corrected that (L711)*

**Response to Interactive comment on by F. Brient on "The Cloud Feedback
Model Intercomparison Project (CFMIP) contribution to CMIP6" by Mark J. Webb et al.**

Reviewer comments below are shown in bold and our responses are in italics.

*Dear Florent,*

**This paper summarizes the objectives of CFMIP and the contribution of CFMIP-3 to
CMIP6.  CFMIP helps to explain the spread of cloud feedbacks, adjustments and pro-
cesses across climate models.  This updated contribution goes a step forward and
suggests additional experiments to allow the community to tackle in more detail the
physical reasons underlying dynamical and regional biases seen in climate models.
By proposing experiments that test especially the atmospheric components of climate
models, CFMIP provides a relevant framework to understand and improve cloud pa-
rameterizations and processes which remain the principal sources of surface and at-
mospheric model biases.
First, the authors summarised well how former CFMIP/CMIP5 experiments helped to
improve our scientific understanding of climate feedbacks.  It thus provides a rele-
vant background supporting the additional experiments that they advise the modelling
groups to perform. I particularly appreciated (1) the will to promote the analysis of ex-
periments when cloud radiative effects are switched off, (2) the pertinent time slice ex-
periments aiming to understand regional climate responses and (3) the encouragement
of a more extensive distribution and use of physical tendencies which are a signature
of the atmospheric components of climate models.**

*Thank you for your careful consideration of our manuscript and for these helpful comments.*

**Below, I have listed a number of minor points which might be addressed to clarify the
text (if the authors find them useful)**

**- Some acronyms are not defined :  AOGCM (l.83), GCM (l.92), RFMIP (l.378), TOA
(l.637), PMC (l.777)**

*We have defined these in the revised manuscript (L92, L101, L494, L802, L985.)*

**L. 196 : I have trouble understanding the meaning of "known answer".**

*We have modified this sentence to read( L255):*

*"Aqua-planet simulations (and other idealized) experiments are particularly effective at highlighting
model differences, for instance in the placement of the tropical rain bands, or in the representation of
cloud changes with warming, as it is not possible to tune them to observations in the same way as is
for more realistic configurations (e.g., Stevens and Bony, 2013)."*

**L. 217:  The amip-future4K experiments used the CMIP3 pattern of SST increase.  Is this pattern consistent with the one derived from CMIP5 models?**

*We haven't looked into this, because we consider consistency with the CMIP5 protocol to be more important than using SSTs from CMIP5 rather than CMIP3.*

*We have added the following to Appendix C (L1043)*

*"We have retained the SST forcing based on the CMIP3 coupled models because we consider it more important to be able to compare CMIP5 and CMIP6 models forced with the same SST pattern than to use a pattern which is consistent with, say, the CMIP5 coupled response."*

**L. 222-225 and L. 419-422: I'm a little bit confused about all 4xCO2 experiments. The amip4xCO2 experiment involves the CO2 effect on the atmospheric component and land warming without the vegetation feedback.  It is thus "equivalent" to the piSST-4xCO2-rad experiment listed in section 2.7 (but not to piSST-4xCO2).  I guess abrupt4xCO2 takes into account the vegetation feedback. So, the amip4xCO2 experiment should be named amip4xCO2-rad, doesn't it?**

*We agree that this would be a more consistent naming of this experiment.  However, we think that the experiment descriptions are clear.  Unfortunately however we understand that CMIP6 experiment names have now been finalised and propagated to the ESG and so it is not now possible to change them.*

**L.257-264: You could also add the reference "Block and Mauristen (13) JAMES - Forcing and Feedback in the MPI-ESM-LR coupled model under abruptly quadrupled CO2", which highlights the utility of diverse amip-pXk and abrupt2xCO2 experiments. L. 288-299:**

*We have added a citation to this paper in section 2.5. (L479)*

**(1) It is thus right that LW effects are the most important contributor to cloud atmospheric radiative effects, and SW effects play a minor role (e.g. Takahashi 09). Nevertheless, local SW cloud effects exist (Pendergrass and Hartmann, 14). It might thus be interesting to point this fact out in the text and leave the discussion about SW effects sufficiently open.**

*We have added the following to section 2.3 (L413)*

*"We note that the presence of clouds does affect the shortwave radiative heating of the atmosphere, although this is a much smaller effect than its longwave equivalent (e.g. Pendergrass and Hartmann, 2014)."*

**(2) Since only LW radiative effects are removed, does it mean that models still have a SW cloud feedback but no LW cloud feedback?**

*Yes. We have clarified this by adding the following to section 2.3 (L413):*

*"In this configuration, the models will have a shortwave cloud feedback but no longwave cloud feedback."*

**(3) "and the radiation code only". Does this mean that, for instance, a boundary-layer parameterization based on LW cloud-top radiative cooling continues to see LW effects?**

*We have added the following comment to section 2.3 (L418):*

*"Care should also be taken to remove the effects of cloud on any longwave cooling used in other model schemes (e.g. turbulent mixing) if these are calculated independently of the radiation scheme. "*

**L.326-328: Contrary to CO2 effects, the radiative forcing of solar insolation depends on latitude. Is this dependency taken into account when the authors state that a 4% change results in a "radiative forcing of a similar magnitude to that due to CO2 quadrupling"?**

*Yes this has been taken into account. We have modified the text as follows to make it clear that this gives a similar magnitude in global mean forcing (L448)*

*"...resulting in a global mean radiative forcing of a similar magnitude to that due to $CO_2$ quadrupling."*

**L. 482: Single Column Model already defined line 91-92.**

*Duplication removed.*

**L. 600-601: Is it normal that "cfDay-2d" is named by CMIP5 and not CFMIP? Why is there no CMIP5 or CFMIP prefix for "cfDay-3d"?**

*The different prefixes represent detail in the formal data request which is not required here. In the manuscript We have deleted the prefixes to avoid confusion, and will add the following sentence (L782-802)*

*"(Please note that in the full data request these variable groups are in many cases split into a number of sub-tables. As noted above, the formal data request provides the definitive specification of the model outputs.)"*

**Fig.1 : The DECK is written in the caption but not highlighted in the graph.**

*We have updated the figure and caption to be consistent in this regard (See Figure 1 and L1052)*

**Fig.1: I consider lwoff experiments as part of the "Clouds" analysis. You may consider making the arrow longer.**

*We have done this. (See Figure 1)*

**Response to Interactive comment on by Anonymous Referee #1 on "The Cloud Feedback Model Intercomparison Project (CFMIP) contribution to CMIP6" by Mark J. Webb et al.**

Reviewer comments below are shown in bold and our responses are in italics.

*Dear Referee,*

**In this paper, the authors state the goals and motivation of CFMIP, review the major accomplishments of previous CFMIPs, and describe the proposed experiments and diagnostics for CFMIP3. The coordinated experiments proposed for CFMIP3 will target a number of outstanding questions for which previous model intercomparisons were not equipped to address, in addition to sustaining a number of highly useful experiments from earlier MIPs that will help to characterize and understand the response of the CMIP6-generation of models to external forcing (in addition to help quantify the forcing itself). Advanced diagnostics (e.g., satellite simulators and high frequency tendency terms) will aid in dissecting model results, and the authors have proposed that they be used more broadly (e.g., COSP turned on for longer durations and in more experiments). The emphasis on (mostly) atmosphere-only simulations in CFMIP3 should hopefully make it appealing for modeling centers to take part in several of the experiments despite the high volume of requested diagnostics.**
**The scientific questions to be addressed by CFMIP3 are well articulated and the various proposed experiments seem well designed to address these questions, and will advance the community's knowledge. The presentation of the paper is not particularly concise, not are the figures particularly insightful, but the writing is clear and overall the presentation seems appropriate for a paper proposing a model intercomparison project. Thus, in my opinion the manuscript represents a substantial contribution to modelling science within the scope of Geoscientific Model Development, and I recommend publication following consideration of some minor comments detailed below.**

*Thank you for your careful consideration of our manuscript and for these helpful comments.*

**Specific Comments:**
**\*piSST and a4SST: It is not clear to me whether (a) monthly- and annually-varying SSTs from the relevant 30 years in the piControl run, or (b) a monthly-resolved climatology of SSTs over the relevant 30 years in the piControl run are prescribed in piSST. Same question for a4SST.**

*We have modified the text to read 'monthly- and –annually varying SSTs....' in the descriptions for piSST, a4SST, a4SSTice and a4SSTice-4xCO2. For consistency we also refer to the AMIP SSTs and sea ice in the amip-a4SST-4xCO2 description as monthly- and –annually varying. (L566, L571, L576, L583, L585, L587, L596).*

**\*amip-piForcing: I'm curious whether there was any interest in performing a similar experiment, but with present-day (rather than preindustrial) forcing held fixed. An example application that occurs to me is that a model with large aerosol-cloud interactions would**

**presumably have brighter clouds with smaller droplets downwind of aerosol sources if the forcing were fixed at present-day, and its temperature-mediated changes in clouds might therefore be different than that occurring in an atmosphere with fewer aerosols. Having these two experiments would allow one to explore this effect (and others related to other forcing agents).**

*This is an interesting idea. However, to be recommended for CMIP by CFMIP, we generally require new experiments to have been piloted and ideally written up with at least one GCM previously. If such an experiment can be demonstrated to provide new insights which are relevant to the objectives of CFMIP then we will certainly consider it in the future.*

**\*Given its implications for understanding apparent state- or time-dependent changes in effective climate sensitivity, I was a little surprised to see no experiments designed to explore causes of nonlinearity in the Gregory plot, perhaps using warming experiments in which the SST pattern is fixed in time (with various patterns), similar to those conducted in Andrews et al, J. Climate (2015). Is there a reason for not proposing these, or are these effects already captured in other proposed experiments?**

*We do consider the causes of non-linearity in abrupt4xCO2 experiments to be an important area to be investigated. The experiments in Andrews at al (2015) were based on actual SSTs from individual models. Pilot studies are ongoing to devise future experiments for CFMIP relevant to this question based on SST pattern responses more representative CMIP5 ensemble mean. We plan to organise a pilot intercomparison based on this, although this might initially be arranged informally within CFMIP rather than as part of CFMIP/CMIP6.*

**\*Line 126: should be "**
**...**
**meetings AND international**
**...**
**"**

*We have corrected this (L157)*

**\*Line 426: "a4SST-4xCO2-all" should be "a4SSTice-4xCO2-all". There may be other instances of this; please verify that they are also changed.**

*This was incorrectly named –the correct name is in fact a4SSTice-4xCO2. Now corrected (L587,589).*

**\*Line 512: What is the reason for dispensing with the cloud tendency terms in CFMIP3?**

*We have added the following at L680:*

*"We have dispensed with the cloud water tendency terms because these have been less widely used than the temperature and humidity tendencies."*

**\*Lines 597-608: it is not clear to me why some of these have a CMIP5 prefix, a CFMIP prefix, or no prefix at all (cfDay-3d). Why would a CMIP5 prefix be appropriate at all?**

*The different prefixes represent detail in the formal data request which is not required here.  In the manuscript we have deleted the prefixes to avoid confusion, and will add the following sentence (L782-802)*

*"(Please note that in the full data request these variable groups are in many cases split into a number of sub-tables.  As noted above, the formal data request provides the definitive specification of the model outputs.)"*

**\*Line 611: should be "**
**...**
**for 140 years OF the piControl**
**...**
**"**

*We have corrected that (L796).*

**\*Appendix A: I don't understand what is meant by "Lead coordinator". Is this the person who has "first dibs" on writing papers based on these experiments?   Are interested investigators expected to contact this person to avoid duplicating work that others are doing with output from these experiments?**

*We have added the following to Appendix A (L933)*

*"We plan to scientifically analyze, evaluate and exploit the proposed experiments and diagnostic outputs, and have identified lead coordinators within CFMIP for different aspects of this activity. The lead coordinators are responsible for encouraging analysis of the relevant experiments as broadly as possible across the scientific community. While they may lead some analysis themselves, they do not have any first claim on analysing or publishing the results.  All interested investigators are encouraged to exploit the data from these experiments. While investigators may wish to liaise with the lead coordinators to avoid duplicating work that others are doing, this is not a requirement."*

**\*Figure 1: I think "CMIP6" should be deleted before "historical".**
**If it is supposed to be there, I don't understand why it is only there.**

*This is the correct naming.  Please see Eyring et al. for the justification.*

**It is also unclear to me why the "Clouds"**
**arrow only extends as far as abrupt-0p5xCO2.  I think both the clouds arrow and the circulation and precipitation arrows should include all experiments,  but in that case, what is the point of showing them?**

*In response to a comment from F. Brient, we have extended the cloud arrow to encompass the lwoff experiments. The timeslice experiments in the bottom group are designed to look at circulation and precipitation responses rather than cloud feedbacks. (See updated Figure 1)*

**\*Table 1: should be "This IS a single**
**…**
**"**

*We have corrected that (See Table 1).*

**\*Table 3: Several of the observational datasets end many years ago despite the fact**
**that these satellites are still in orbit. Are there plans to extend these records, especially**
**since the AMIP runs end in 2015?**

*We have added the following the end of Section 3.2 (L814):*

*"These datasets are periodically updated to include more recent data from the relevant satellites, many of which are still operational. Please refer to the CFMIP-OBS website for updates."*

Response to Interactive comment by Anonymous Referee #2 on "The Cloud Feedback Model Intercomparison Project (CFMIP) contribution to CMIP6" by Mark J. Webb et al.

Reviewer comments below are shown in bold and our responses are in italics.

*Dear Referee,*

**This manuscript outlines the CFMIP-3 experimental strategies, the associated model output, and the motivation and anticipated results of these experiments. Overall the manuscript is clearly written and accurately summarizes the plans for CFMIP-3, and in many ways represents more of a review of past CFMIP achievements, which in itself is a useful contribution. I recommend acceptance with only minor revisions as outlined in my suggestions below.**

*Thank you for your careful consideration of our manuscript and for these helpful comments.*

**The authors are very generous in their citations of other work, which is commendable, but it detracts from the readability of the manuscript. I recommend the authors consider focusing on a few select highlights of the previous CFMIPs that illustrate the main contributions, rather than attempting an exhaustive summary of everything that's been learned from CFMIP experiments. In the current form, it's difficult to identify what the key contributions of CFMIP have been.**

*We appreciate that the many citations do make the manuscript difficult to read in places. We are glad that the review of the main CFMIP achievements is appreciated, and agree that this could be achieved with fewer citations. However, we also consider it important to communicate the full breadth of studies arising from CFMIP, as this will we think help to inform the decisions made by modelling groups on which CFMIP experiments to perform and which model outputs to provide. Following guidance in the subsequent interactive comment from the Editor (Julia Hargreaves) we have reduced the number of citations in the introduction, in particular where there is duplication with Section 2. Throughout, where several citations are made together, we have broken them into smaller groups as suggested to give the reader a better idea of what distinguishes them. (See for example L230-239, L248, L673).*

**Section 2.1 reads more as a review of all previous studies that used CFMIP data, rather than a description of the CFMIP-3. I would recommend moving much of this to the previous section which reviews past CFMIPs and identify any changes/deletions from the past CFMIPs before then proceeding to describe the new additions to the CFMIP-3 set of experiments.**

*We appreciate that there is some duplication between the text in the introduction and in Section 2.1, in particular in the case of citations. We have addressed this by modifying the text in the introduction, as described above. We considered the referee's suggestion to move the bulk of this to the introduction, and to then to describe these Tier I experiments in terms of changes/deletions compared to those in previous CFMIPs. However, as pointed out in the subsequent comment by the Editor, it is important that, as a MIP documentation paper, we document the experiments in such a way as to allow a third party could set up each run from the information provided. We think that recapping on the CFMIP-2 experiment protocol in the introduction and then introducing aspects of the CFMIP-3 protocol as changes relative to this would make it harder for modelling groups to use this paper as the definitive specification for the CFMIP-3 experiments, and so prefer to leave the structure as it is presently.*

**It would also be useful to define what a "DECK" is.**

*We have modified the text at L175 to read:*

*"Most of the CFMIP-3 experiments are based on CO2 concentration forced amip, piControl and abrupt-4xCO2 CMIP DECK (Diagnostic, Evaluation and Characterization of Klima) experiments (Eyring et al., 2016)."*

**There is rightfully considerable attention within CFMIP devoted to isolating and quanti- fying the fast adjustments. However the fast adjustments arise from both atmospheric radiative heating changes and land warming.**

*We agree. We checked the manuscript, and all references to tropospheric adjustments do also refer to land warming.*

**It would be useful to isolate these contributions (beyond the use of aqua planets, whose utility in quantifying CGCM feedbacks is a little over sold here IMO). Has there been any efforts to develop experiments for this? If not, this issue might warrant some discussion in reference to the experiments designed to quantify adjustments.**

*We agree that experiments designed to separate the effects of land warming and atmospheric heating in realistic experiments would be useful . However we are not aware of any published studies which demonstrate a way to do this. To be recommended for CMIP by CFMIP, we generally require new experiments to have been piloted and ideally written up with at least one GCM previously. If such an experiment can be demonstrated to provide new insights which are relevant to the objectives of CFMIP then we will certainly consider it in the future.*

**Response to Interactive comment on by A. Voigt on "The Cloud Feedback Model Intercomparison Project (CFMIP) contribution to CMIP6" by Mark J. Webb et al.**

Reviewer comments below are shown in bold and our responses are in italics.

*Dear Aiko,*

**The authors provide a concise and well-written presentation of the CFMIP experiments proposed for CMIP6, which will continue the successful CFMIP activities over the last 15 years. I enjoyed reading the paper, in particular the historical context given in the introduction, and find that it nicely presents the scientific motivation and chosen simulation strategy at a level amenable to both CFMIP experts and climate scientists with other backgrounds. I recommend publication in GMD after my following minor comments have been addressed.**

*Thank you for your careful consideration of our manuscript and for these helpful comments.*

**Line 217, amip-future4K simulations: Why is the CMIP3 SST pattern used and not an updated pattern from CMIP5 AOGCM runs?**

*We have added the following to Appendix C (L1068):*

*"We have retained the SST forcing based on the CMIP3 coupled models because we consider it more important to be able to compare CMIP5 and CMIP6 models forced with the same SST pattern than to use a pattern which is consistent with, say, the CMIP5 coupled response."*

**Line 227: I am very glad to hear that the CMIP-3 aquaplanet simulations will be extended to 10 years. This will be beneficial for studies of extratropical dynamics, for which internal variability is larger than in the tropics.**

*Thank you.*

**Line 238, amip-m4k simulations: I am wondering to what extent some models might have problems with SSTs below freezing? Maybe this might require code changes in some models in case they employ a fixed lower threshold for the SST used in the calculation of surface fluxes? Such a problem would, of course, not occur for the p4K simulations?**

*We have added the following at L356:*

*"In models which employ a fixed lower threshold near freezing for the SST used in the calculation of the surface fluxes, this should ideally also be reduced by 4K."*

**Lines 279: The authors might consider to also refer to Voigt and Shaw (2015, Nature Geoscience) here for the extratropical circulation. The study showed that cloud-**

**radiative feedbacks contribute substantially to the poleward jet shifts under 4K warming in aquaplanet simulations.**

*We have added that reference at L379.*

**Line 266, lwoff experiments: Just an idea, but I though it's worthwhile bringing it up here: While the surface cloud effect is stronger in the shortwave than the longwave domain, the longwave can still substantial. I am wondering whether an experiment with clear-sky heating in the atmosphere and all-sky heating at the surface would be even better to isolate the effect of atmospheric cloud-radiative heating. I suspect it's too late to change the experimental protocol, and maybe there is a reason why lwoff is still better. If so, it might be worthwhile to briefly discuss this.**

*This is an interesting idea thank you. However, to be recommended for CMIP by CFMIP, we generally require new experiments to have been piloted and ideally written up with at least one GCM previously. The lwoff are experiments currently proposed are very similar to those piloted by Fermepin and Bony, 2014, and technically easier to implement than what is proposed. If such an experiment can be demonstrated to provide new insights which are relevant to the objectives of CFMIP then we will certainly consider it in the future.*

*We have added the following to the manuscript at L417:*

*"An alternative method (proposed by A. Voigt) was also considered, in which clear-sky heating rates would be applied in the atmosphere while retaining the all-sky fluxes at the surface. Although this approach would potentially isolate the effects of cloud heating in the atmosphere more cleanly than the lwoff experiments proposed here, it is yet to be demonstrated in a pilot study, and is considered more technically difficult to implement than the lwoff experiments, which are very similar to those piloted by Fermepin and Bony, 2014."*

**Line 342: Non-linearity was also shown in the CMIP5 ensemble by Meraner et al. (2013, GRL, doi:10.1002/2013GL058118). Meraner et al. showed non-linear climate sensitivity across the multi-model CMIP5 ensemble, whereas the other cited work used single models if I am not mistaken. So maybe worthwhile including here?**

*We have added that reference at L455:*

**Line 368: Maybe specify the reason why the CFMIP2/CMIP5 runs did not allow such an estimate. I.e., I assume that one would use SST-driven simulations for this and that the usual amip period is too short to reliably calculate feedbacks?**

*We have modified line to read (L485):*

*"The previous CFMIP-2/CMIP5 design was unable to diagnose the time-variation of feedbacks of explicit relevance to the historical period, because this requires the removal of the time varying forcing."*

**Sect. 2.7: The time slice experiments ask an interesting question but given that 8 experiments are demanded, I was wondering how they ought to be combined to answer the questions in mind. Maybe the authors can give an example?**

*We have added two examples in the text of how these experiments are combined at L576:*

*"The time slice experiments can be combined in various ways to isolate the climate response to each individual aspect of forcing and warming. For example the response to SST pattern change is given by taking the difference between a4SST and piSST-pxK, and the plant physiological response is found by taking the difference between piSST-4xCO2 and piSST-4xCO2-rad."*

**Line 487: I would be curious to know about the reasons to no longer ask for cfSites output in the aquaplanet ensemble and amip-future4K. Is it the lack of observational data to compare to, or a choice to avoid asking for too much data?**

*We have amended the manuscript at L640 to read:*

*"We have dispensed with the cfSites outputs in the aquaplanet and amip-future4K experiments because these have been less widely used compared to those from the other experiments."*

**Line 685: and –> an**

*We have amended that.*

**Figure 1: Why does the vertical cloud bar on the right side not include the lwoff simulations?**

*This point was also raised by F. Brient. We have extended the arrow to include the lwoff experiment (See Figure 1).*

**For some of the proposed simulations the link to clouds, which are the prime motivation for CFMIP, is not very evident and maybe could be made clearer? I am thinking of the simulations in Sect. 2.7 (time slice experiments) and Sect. 2.5 (nonLinMIP).**

The primary objective of CFMIP is to inform future assessments of cloud feedbacks through improved understanding of cloud-climate feedback mechanisms and better evaluation of cloud processes and cloud feedbacks in climate models. However, the CFMIP approach is also increasingly being used to understand other aspects of climate change, and so a second objective has now been introduced, to improve understanding of circulation, regional-scale precipitation, and non-linear changes. For this reason, not all experiments need to be relevant to clouds. We have modified the text in the abstract, introduction and conclusions to state this explicitly (L26,L127,L849).

**Response to Interactive comment by J. C. Hargreaves (Editor) on "The Cloud Feedback Model Intercomparison Project (CFMIP) contribution to CMIP6" by Mark J. Webb et al.**

Editor comments below are shown in bold and our responses are in italics.

*Dear Julia,*

**This seems to me to be a pretty good MIP manuscript. You do have the advantage that the protocols for the experiments are relatively simple to describe, but I still think it very well organised.**

*Thank you.*

**I haven't checked all the protocols - just a couple that I am particularly interested in - but the information seemed complete for those. However, please do check through your revised manuscript to make sure that a third party could set up each run from the information provided.**

*We have done this, and have made a few changes to clarify some issues:*

*We have updated the amip-future4K experiment definition at L274 to include the sentence:*

[revised manuscript text omitted]

**The remaining peculiarity is the reference to boundary conditions that will become available through other papers in this special issue - are there now references that can be provided for these papers?**

*We have added the following at L176:*

*"Most of the CFMIP-3 experiments are based on $CO_2$ concentration forced amip, piControl and abrupt-4xCO2 CMIP DECK (Diagnostic, Evaluation  and  Characterization  of  Klima) experiments (Eyring et al., 2016).  Unless otherwise specified below, the CFMIP-3 experiments should be configured consistently with the DECK experiments on which they are based, using consistent model formulation, and forcings and boundary conditions as specified by Eyring et al., 2016."*

**The thing I spotted in the reviewers' comments that I am unsure about is the suggestion**

**to abbreviate the citations to increase readability. Here's an example,**
**"Temperature and humidity tendency terms in particular have been shown to be useful**
**for understanding the roles of different parts of the model physics in cloud feedbacks**
**and adjustments (Kamae and Watanabe 2012; Williams et al., 2013; Webb and 509**
**Lock 2013; Demoto et al., 2013; Sherwood et al., 2014; Ogura et al., 2014; Brient et**
**al., 2015)"**
**I generally don't like the idea of reducing the citations, but that is an awful lot of refer-**
**ences all apparently showing the same thing!  As a reader I'd want to know what the**
**difference is between these papers, and which one I should look up in order to learn**
**about the thing I am specifically interested in.  The obvious solution would be to add**
**a little more description, so that the reader has more knowledge about the content of**
**the references. Doing so will make the manuscript longer, which could get out of hand,**
**but maybe there is a middle way which produces a more readable and more useful**
**manuscript.**

*We have addressed this in the manuscript as follows.  We have reduced the number of citations in*
*the introduction, in particular where there is duplication with Section 2. Throughout, where several*
*citations are made together, we have broken  them into smaller groups as suggested to give the*
*reader a better idea of what distinguishes them.  Please see L55-75, L105-113, L233-240, L249-250,*
*L656-657.*

*For example, in the case highlighted above, we have updated the manuscript at L656 to read:*

[revised manuscript text omitted]

2006.

**CFMIP-GMD-Paper-161012**

| Main document changes and comments | | |
|---|---|---|
| **Page 1: Deleted** | **mark.webb** | **28/09/2016 09:34:00** |

Submitted

| **Page 1: Inserted** | **mark.webb** | **28/09/2016 09:34:00** |
|---|---|---|

Revised for

| **Page 1: Deleted** | **mark.webb** | **28/09/2016 09:34:00** |
|---|---|---|

to

| **Page 1: Inserted** | **mark.webb** | **12/10/2016 09:50:00** |
|---|---|---|

12[th] October

| **Page 1: Formatted** | **mark.webb** | **12/10/2016 09:50:00** |
|---|---|---|

Superscript

| **Page 1: Deleted** | **mark.webb** | **28/09/2016 09:34:00** |
|---|---|---|

30[th] March

| **Page 1: Inserted** | **mark.webb** | **31/08/2016 13:34:00** |
|---|---|---|

the CFMIP approach is also increasingly being used to understand other aspects of climate change, and so a second objective has now been introduced, to improve understanding of circulation, regional-scale precipitation, and non-linear changes.

| **Page 1: Deleted** | **mark.webb** | **31/08/2016 13:34:00** |
|---|---|---|

the CFMIP approach is also increasingly being used to understand other aspects of climate change, such as nonlinear change and regional changes in atmospheric circulation and precipitation.

| **Page 1: Deleted** | **mark.webb** | **30/08/2016 14:06:00** |
|---|---|---|

; Vial et al., 2013

| **Page 2: Inserted** | **mark.webb** | **30/08/2016 14:06:00** |
|---|---|---|

, both in fine resolution models

| **Page 2: Deleted** | **mark.webb** | **30/08/2016 14:22:00** |
|---|---|---|

| **Page 2: Inserted** | **mark.webb** | **30/08/2016 14:07:00** |
|---|---|---|

 Bretherton et al., 2015) and in comprehensive climate models (e.g.

| **Page 2: Deleted** | **mark.webb** | **30/08/2016 14:08:00** |
|---|---|---|

; Webb and Lock 2013; Brient and Bony 2013

| **Page 2: Deleted** | **mark.webb** | **30/08/2016 14:08:00** |
|---|---|---|

Ringer et al., 2014; Medeiros et al., 2015; Bretherton et al., 2015;

| **Page 2: Inserted** | **mark.webb** | **12/10/2016 10:13:00** |
|---|---|---|

; Andrews and Gregory, 2016

| **Page 2: Deleted** | **mark.webb** | **30/08/2016 14:12:00** |
|---|---|---|

; Andrews et al., 2015

| **Page 2: Deleted** | **mark.webb** | **30/08/2016 14:22:00** |
|---|---|---|

,

| **Page 2: Deleted** | **mark.webb** | **30/08/2016 15:25:00** |
|---|---|---|

 Brient et al., 2015; Tsushima et al., 2015;

| **Page 2: Inserted** | **mark.webb** | **15/09/2016 15:24:00** |
|---|---|---|

,

| Page 2: Deleted | mark.webb | 30/08/2016 14:23:00 |

Marchand and Ackerman, 2010;

| Page 2: Deleted | mark.webb | 30/08/2016 14:28:00 |

Nam et al., 2012a;

| Page 2: Deleted | mark.webb | 30/08/2016 14:28:00 |

; Klein et al., 2013; Chepfer et al., 2014

| Page 2: Inserted | mark.webb | 25/08/2016 10:12:00 |

(atmosphere–ocean general circulation model)

| Page 2: Deleted | mark.webb | 25/08/2016 10:17:00 |

s

| Page 2: Inserted | mark.webb | 25/08/2016 10:18:00 |

General Circulation Models (

| Page 2: Inserted | mark.webb | 25/08/2016 10:17:00 |

)

| Page 2: Deleted | mark.webb | 10/10/2016 17:10:00 |

| Page 2: Deleted | mark.webb | 30/08/2016 14:32:00 |

Nam et al., 2012; Gregory and Chepfer, 2012;

| Page 2: Inserted | mark.webb | 30/08/2016 14:31:00 |

, Chepfer et al., 2014

| Page 2: Deleted | mark.webb | 30/08/2016 14:44:00 |

Webb and Lock, 2013;

| Page 2: Inserted | mark.webb | 15/09/2016 15:31:00 |

    The primary objective of CFMIP is to inform future assessments of cloud feedbacks through improved understanding of cloud-climate feedback mechanisms and better evaluation of cloud processes and cloud feedbacks in climate models. However, the CFMIP approach is also increasingly being used to understand other aspects of climate change, and so a second objective has been introduced, to improve understanding of circulation, regional-scale precipitation, and non-linear changes.

| Page 3: Deleted | mark.webb | 31/08/2016 13:30:00 |

The primary goal of CFMIP is to inform improved assessments of cloud feedbacks on climate change.  However, the CFMIP approach is increasingly being used to understand other aspects of climate response, such as regional circulation and precipitation changes, and non-linear changes.

| Page 3: Inserted | mark.webb | 15/09/2016 15:31:00 |

| Page 3: Inserted | mark.webb | 26/08/2016 13:23:00 |

and

[revised manuscript text omitted]

- and annually-

| Page 8: Deleted | mark.webb | 26/08/2016 12:54:00 |

-

| Page 8: Inserted | mark.webb | 01/09/2016 14:17:00 |

and annually-

| Page 8: Inserted | mark.webb | 24/05/2016 14:39:00 |

open

| Page 8: Inserted | mark.webb | 26/08/2016 12:56:00 |

and annually-

| Page 8: Inserted | mark.webb | 26/08/2016 12:57:00 |

and annually-

| Page 8: Deleted | mark.webb | 20/05/2016 15:03:00 |

*a4SST-4xCO2*

| Page 8: Inserted | mark.webb | 20/05/2016 15:03:00 |

*a4SSTice-4xCO2*

| Page 8: Inserted | mark.webb | 26/08/2016 12:58:00 |

and annually-

| Page 8: Deleted | mark.webb | 20/05/2016 15:02:00 |
|---|---|---|

*a4SST-4xCO2*

| Page 8: Inserted | mark.webb | 20/05/2016 15:02:00 |
|---|---|---|

*a4SSTice-4xCO2*

| Page 8: Inserted | mark.webb | 01/09/2016 14:15:00 |
|---|---|---|

 The time slice experiments can be combined in various ways to isolate the climate response to each individual aspect of forcing and warming. For example the response to SST pattern change is given by taking the difference between  *a4SST* and *piSST-pxK,* and the plant physiological response is found by taking the difference between *piSST-4xCO2* and *piSST-4xCO2-rad*.

| Page 8: Inserted | mark.webb | 26/08/2016 13:00:00 |
|---|---|---|

 and annually-

| Page 8: Deleted | mark.webb | 24/08/2016 14:02:00 |
|---|---|---|

| Page 8: Deleted | mark.webb | 20/05/2016 15:03:00 |
|---|---|---|

*a4SST-4xCO2*

| Page 8: Inserted | mark.webb | 20/05/2016 15:03:00 |
|---|---|---|

*a4SSTice-4xCO2*

| Page 8: Deleted | mark.webb | 01/09/2016 14:18:00 |
|---|---|---|

| Page 8: Inserted | mark.webb | 27/09/2016 11:57:00 |
|---|---|---|

https://www.earthsystemcog.org/projects/wip/CMIP6DataRequest

| Page 9: Deleted | mark.webb | 25/08/2016 14:06:00 |
|---|---|---|

ingle

| Page 9: Deleted | mark.webb | 25/08/2016 14:06:00 |
|---|---|---|

olumn

| Page 9: Deleted | mark.webb | 25/08/2016 14:06:00 |
|---|---|---|

odels

| Page 9: Deleted | mark.webb | 26/08/2016 13:28:00 |
|---|---|---|

we have dispensed with the cfSites outputs in the aquaplanet experiments and in *amip-future4K*.

| Page 9: Inserted | mark.webb | 12/10/2016 10:25:00 |
|---|---|---|

| Page 9: Deleted | mark.webb | 12/10/2016 10:25:00 |
|---|---|---|

| Page 9: Inserted | mark.webb | 26/08/2016 13:28:00 |
|---|---|---|

We have dispensed with the cfSites outputs in the aquaplanet and *amip-future4K* experiments because these have been less widely used compared to those from the other experiments.

| Page 9: Deleted | mark.webb | 30/08/2016 15:33:00 |
|---|---|---|

and adjustments

| Page 9: Inserted | mark.webb | 30/08/2016 15:33:00 |
|---|---|---|

e.g.

| Page 9: Deleted | mark.webb | 30/08/2016 15:33:00 |
|---|---|---|

Kamae and Watanabe 2012;

| Page 9: Moved to page 9 (Move #3) | mark.webb | 30/08/2016 15:35:00 |
|---|---|---|

Williams et al., 2013;

| Page 9: Deleted | mark.webb | 30/08/2016 15:35:00 |
|---|---|---|

 Ogura et al., 2014;

| Page 9: Inserted | mark.webb | 30/08/2016 15:33:00 |
|---|---|---|

and cloud adjustments (e.g. Kamae and Watanabe 2012; Ogura et al., 2014)

| Page 9: Moved from page 9 (Move #3) | mark.webb | 30/08/2016 15:35:00 |
|---|---|---|

Williams et al., 2013;

| Page 9: Deleted | mark.webb | 31/08/2016 13:13:00 |
|---|---|---|

dispensed with the cloud tendency terms,

| Page 9: Deleted | mark.webb | 31/08/2016 13:14:00 |
|---|---|---|

| Page 9: Inserted | mark.webb | 31/08/2016 13:12:00 |
|---|---|---|

We have dispensed with the cloud water tendency terms because these have been less widely used than the temperature and humidity tendencies.

| Page 10: Inserted | mark.webb | 24/08/2016 14:07:00 |
|---|---|---|

 of

| Page 10: Inserted | mark.webb | 27/09/2016 11:58:00 |
|---|---|---|

https://www.earthsystemcog.org/projects/wip/CMIP6DataRequest

| Page 11: Deleted | mark.webb | 10/10/2016 17:11:00 |
|---|---|---|

web site

| Page 11: Inserted | mark.webb | 10/10/2016 17:11:00 |
|---|---|---|

website

| Page 11: Deleted | mark.webb | 25/08/2016 14:21:00 |
|---|---|---|

CFMIP-

| Page 11: Inserted | mark.webb | 25/08/2016 14:21:00 |
|---|---|---|

–

| Page 11: Deleted | mark.webb | 25/08/2016 14:21:00 |
|---|---|---|

-

| Page 11: Deleted | mark.webb | 25/08/2016 14:21:00 |
|---|---|---|

CMIP5-

| Page 11: Inserted | mark.webb | 25/08/2016 14:21:00 |
|---|---|---|

–

| Page 11: Deleted | mark.webb | 25/08/2016 14:21:00 |
|---|---|---|

-

| Page 11: Inserted | mark.webb | 25/08/2016 14:21:00 |
|---|---|---|

–

| Page 11: Deleted | mark.webb | 25/08/2016 14:21:00 |
|---|---|---|

-

| Page 11: Deleted | mark.webb | 25/08/2016 14:22:00 |
|---|---|---|

CFMIP-

| Page 11: Deleted | mark.webb | 25/08/2016 14:23:00 |
|---|---|---|

CFMIP-

| **Page 11: Deleted** | **mark.webb** | **25/08/2016 14:23:00** |

CFMIP-

| **Page 11: Deleted** | **mark.webb** | **25/08/2016 14:23:00** |

CFMIP-

| **Page 11: Inserted** | **mark.webb** | **25/08/2016 14:24:00** |

_

| **Page 11: Deleted** | **mark.webb** | **25/08/2016 14:24:00** |

-

| **Page 11: Deleted** | **mark.webb** | **25/08/2016 14:24:00** |

CMIP5-

| **Page 11: Inserted** | **mark.webb** | **25/08/2016 14:24:00** |

_

| **Page 11: Deleted** | **mark.webb** | **25/08/2016 14:24:00** |

-

| **Page 11: Inserted** | **mark.webb** | **26/08/2016 13:34:00** |

of

| **Page 11: Inserted** | **mark.webb** | **25/08/2016 14:24:00** |

_

| **Page 11: Deleted** | **mark.webb** | **25/08/2016 14:24:00** |

-

| **Page 11: Deleted** | **mark.webb** | **25/08/2016 14:25:00** |

CFMIP-

| **Page 11: Deleted** | **mark.webb** | **25/08/2016 14:25:00** |

CFMIP-

| **Page 11: Deleted** | **mark.webb** | **25/08/2016 14:24:00** |

CFMIP-

[revised manuscript text omitted]

O.

| Page 22: Inserted | mark.webb | 20/05/2016 15:00:00 |
|---|---|---|

O.,

| Page 22: Deleted | mark.webb | 20/05/2016 15:00:00 |
|---|---|---|

D.

| Page 22: Inserted | mark.webb | 20/05/2016 15:00:00 |
|---|---|---|

D.,

| Page 22: Deleted | mark.webb | 20/05/2016 15:00:00 |
|---|---|---|

G.

| Page 22: Inserted | mark.webb | 20/05/2016 15:00:00 |
|---|---|---|

G.,

| Page 22: Deleted | mark.webb | 20/05/2016 15:00:00 |
|---|---|---|

A.

| Page 22: Inserted | mark.webb | 20/05/2016 15:00:00 |
|---|---|---|

A. ,

| Page 22: Moved to page 22 (Move #1) | mark.webb | 20/05/2016 15:01:00 |
|---|---|---|

D. J. L.

| Page 22: Moved from page 22 (Move #1) | mark.webb | 20/05/2016 15:01:00 |
|---|---|---|

D. J. L.

| Page 22: Deleted | mark.webb | 20/05/2016 15:01:00 |
|---|---|---|

S.

| Page 22: Inserted | mark.webb | 20/05/2016 15:01:00 |

, S.

| Page 22: Inserted | mark.webb | 25/08/2016 13:35:00 |

Good, P., Andrews, T., Chadwick, R., Dufresne, J. L., Gregory, J. M., Lowe, J. A., Schaller, N., and Shiogama, H.: The nonlinMIP intercomparison project: physical basis, experimental design and analysis principles, Geosci. Model Dev. Discuss., doi:10.5194/gmd-2016-56, in review, 2016.

| Page 22: Deleted | mark.webb | 25/08/2016 13:38:00 |

Good, P., Andrews, T., Chadwick, R., Dufresne, J. L., Gregory, J. M., Lowe, J. A., Schaller, N., Shiogama, H. : The nonlinMIP intercomparison project: physical basis, experimental design and analysis principles.  Geophys Model Dev., submitted.

| Page 22: Inserted | mark.webb | 25/08/2016 13:36:00 |

Gregory, J. M., and Andrews, T.: Variation in climate sensitivity and feedback parameters during the historical period, Geophys. Res. Lett., 43, 3911–3920, doi:10.1002/2016GL068406, 2016.

| Page 22: Deleted | mark.webb | 25/08/2016 13:36:00 |

Gregory, J.M., and Andrews, T.: Variation in climate sensitivity and feedback during the historical period. Submitted to Geophysical Research Letters.

| Page 23: Inserted | mark.webb | 25/08/2016 10:24:00 |

Kageyama, M., Braconnot, P., Harrison, S. P., Haywood, A. M., Jungclaus, J., Otto-Bliesner, B. L., Peterschmitt, J.-Y., Abe-Ouchi, A., Albani, S., Bartlein, P. J., Brierley, C., Crucifix, M., Dolan, A., Fernandez-Donado, L., Fischer, H., Hopcroft, P. O., Ivanovic, R. F., Lambert, F., Lunt, D. J., Mahowald, N. M., Peltier, W. R., Phipps, S. J., Roche, D. M., Schmidt, G. A., Tarasov, L., Valdes, P. J., Zhang, Q., and Zhou, T.: PMIP4-CMIP6: the contribution of the Paleoclimate Modelling Intercomparison Project to CMIP6, Geosci. Model Dev. Discuss., doi:10.5194/gmd-2016-106, in review, 2016.

| Page 24: Inserted | mark.webb | 31/08/2016 11:59:00 |

Meraner, K., Mauritsen, T. and Voigt, A.: Robust increase in equilibrium climate sensitivity under global warming. Geophysical Research Letters, 40(22), pp.5944-5948, 2013.

| Page 24: Deleted | mark.webb | 30/08/2016 14:35:00 |

a

| Page 24: Deleted | mark.webb | 30/08/2016 14:35:00 |

Nam, C. C. W., and Quaas, J.: Evaluation of Clouds and Precipitation in the ECHAM5 General Circulation Model Using CALIPSO and CloudSat Satellite Data. I,  J. Climate, 25, 4975-4992. DOI:10.1175/JCLI-D-11-00347.1, 2012b.

| Page 24: Inserted | mark.webb | 25/08/2016 13:51:00 |

Pendergrass, A.G. and Hartmann, D.L.: The atmospheric energy constraint on global-mean precipitation change. Journal of Climate, 27(2), pp.757-768, 2014.

| Page 25: Inserted | mark.webb | 25/08/2016 10:26:00 |

Pincus, R., Forster, P. M., and Stevens, B.: The Radiative Forcing Model Intercomparison Project (RFMIP): Experimental Protocol for CMIP6, Geosci. Model Dev. Discuss., doi:10.5194/gmd-2016-88, in review, 2016.

**Header and footer changes**

**Text Box changes**

**Header and footer text box changes**

**Footnote changes**

**Endnote changes**

---

## Author Response (AR2)

**Response to Topical Editor Decision: Publish subject to minor revisions (Editor review)**
**by J. C. Hargreaves on "The Cloud Feedback Model Intercomparison Project (CFMIP) contribution to CMIP6" by Mark J. Webb et al.**

Editor comments below are shown in bold and our responses are in italics.

*Dear Julia,*

**Thanks for the revised manuscript and thorough responses to all the reviews and comments. I just have a few mostly technical questions.**

*Thank you.*

**The nomenclature in regards to the whole project is a bit inconsistent. It is stated that "CFMIP-3" is, in this paper, to apply to the CFMIP-3 in CMIP6 runs, while also acknowledging that there will be other CFMIP-3 runs that are not included in CMIP6. What are they to be called? At the same time, throughout the paper reference is made to "CMIP5/CFMIP-2", which presumably refers to the subset of CFMIP-2 runs that were included in CMIP5. Therefore I think it might be more consistent to use "CMIP6/CFMIP-3" in this manuscript, particularly in places where a comparison is being made with "CMIP5/CFMIP-2".**

*We agree that this is inconsistent. We think the best way to address this is to refer to the broader CFMIP-3 project as CFMIP-3, but to refer to the various experiments as 'CFMIP-2/CMIP5 experiments', ' CFMIP-3/CMIP6 experiments' and 'informal CFMIP-3 experiments'. We have amended the manuscript throughout. We have also rewritten the relevant text in the introduction thus:*

*" CFMIP is now entering its third phase, CFMIP-3, which will run in parallel with the current phase of the Coupled Model Intercomparison Project (CMIP6, Eyring et al., 2016) This paper documents the CFMIP-3/CMIP6 experiments and diagnostic outputs which constitute the CFMIP-3 contribution to CMIP6. It is anticipated that CFMIP-3 will be broader than what is described here, for instance including studies with process models, and informal CFMIP-3 experiments which are organised independently of CMIP6. Please refer to the CFMIP website for announcements of these other initiatives and CFMIP annual meetings."*

**The nonlinearity of the Gregory plot is alluded to by Anonymous reviewer 1, and you responded, but do not include this discussion in the manuscript. It seems to me that it should be added, as otherwise the paper may give many people the impression that the estimation of climate sensitivity by the Gregory method is wholly accurate.**

*We have added the following to the end of Section 2.7:*

*"We also consider the time variation of feedbacks in abrupt-4xCO2 experiments to be an important area to be investigated, as this can have a substantial impact on estimates of equilibrium sensitivity*

*(e.g. Geoffroy et al., 2013).   Andrews et al., 2015 investigated such effects using two atmosphere-only GCMs forced with SSTs and sea ice from their own abrupt-4xCO2 experiments, and attributed the time variation in the feedbacks to changes in the pattern of surface warming.  Pilot studies are ongoing to develop similar experiments based on a composite SST pattern response more representative if the CMIP5 ensemble mean.  We plan to organise an informal pilot intercomparison based on this within CFMIP-3 and may subsequently propose these experiments as an extension to the CFMIP-3/CMIP6 experiment set."*

**It seems a bit inconsistent that discussion of Aiko's suggestion for an experiment has been added to the paper, but the suggestions made by Anonymous reviewers 1 and 2 have not been mentioned. What is the reason for this?**

*Anonymous reviewers 1 and 2 both proposed new experiments, but Aiko proposed doing existing experiments slightly differently.  We felt that it was appropriate to mention Aiko's points in the text because they are relevant to the design of the experiments which we \*are\* proposing for CFMIP-3/CMIP6, as opposed to the suggestions from anonymous reviewers which are about experiments which we \*aren't\* proposing.*

**It is a shame that it is not anymore possible to change the names of the individual runs as, if one reviewer is confused by the naming, then other people will be too.**

*Yes, we agree this is unfortunate.  However the description of the experiment is clear.  To minimise potential confusion we have added the following to the manuscript:*

[revised manuscript text omitted]

2006.

**CFMIP-GMD-Paper-161028**

| Main document changes and comments |
|---|

| Page 1: Inserted | mark.webb | 27/10/2016 10:02:00 |
|---|---|---|

| Page 1: Deleted | mark.webb | 27/10/2016 10:02:00 |
|---|---|---|

| Page 3: Inserted | mark.webb | 28/10/2016 15:34:00 |
|---|---|---|

CFMIP is now entering its third phase, CFMIP-3, which will run in parallel with the current phase of the Coupled Model Intercomparison Project (CMIP6, Eyring et al., 2016)

| Page 3: Deleted | mark.webb | 27/10/2016 10:20:00 |
|---|---|---|

describes and

| Page 3: Deleted | mark.webb | 27/10/2016 10:20:00 |
|---|---|---|

| Page 3: Inserted | mark.webb | 27/10/2016 10:20:00 |
|---|---|---|

 the CFMIP-3/CMIP6 experiments and diagnostic outputs which constitute the CFMIP-3 contribution to CMIP6.

| Page 3: Deleted | mark.webb | 27/10/2016 10:20:00 |
|---|---|---|

the CFMIP contribution to the current phase on the Coupled Model Intercomparison Project (CMIP6, Eyring et al., 2016).

| Page 3: Deleted | mark.webb | 28/10/2016 15:40:00 |
|---|---|---|

 eventually

| Page 3: Inserted | mark.webb | 27/10/2016 10:17:00 |
|---|---|---|

and informal CFMIP-3 experiments which are organised independently of CMIP6.  Please refer to the CFMIP website for announcements of these other initiatives and  CFMIP annual meetings.

| Page 3: Deleted | mark.webb | 27/10/2016 10:41:00 |
|---|---|---|

but for the purposes of this document CFMIP-3 should be considered to be synonymous with the CFMIP contribution to CMIP6.

| Page 3: Inserted | mark.webb | 27/10/2016 10:29:00 |
|---|---|---|

/CMIP6

| Page 3: Deleted | mark.webb | 27/10/2016 10:29:00 |
|---|---|---|

proposed for CMIP6

| Page 3: Inserted | mark.webb | 27/10/2016 10:41:00 |
|---|---|---|

/CMIP6

| Page 3: Inserted | mark.webb | 27/10/2016 10:42:00 |
|---|---|---|

/CMIP6

| Page 3: Inserted | mark.webb | 27/10/2016 10:42:00 |
|---|---|---|

/CMIP6

| Page 3: Deleted | mark.webb | 27/10/2016 10:30:00 |
|---|---|---|

CFMIP-3

| Page 3: Inserted | mark.webb | 27/10/2016 10:30:00 |
|---|---|---|

CFMIP-3/CMIP6

| Page 3: Inserted | mark.webb | 27/10/2016 10:43:00 |
|---|---|---|

-3

| Page 3: Deleted | mark.webb | 27/10/2016 10:30:00 |
|---|---|---|

CFMIP-3

| Page 3: Inserted | mark.webb | 27/10/2016 10:30:00 |
|---|---|---|

CFMIP-3/CMIP6

| Page 3: Deleted | mark.webb | 27/10/2016 10:30:00 |
|---|---|---|

**CFMIP-3**

| Page 3: Inserted | mark.webb | 27/10/2016 10:30:00 |
|---|---|---|

**CFMIP-3/CMIP6**

| Page 3: Deleted | mark.webb | 27/10/2016 10:08:00 |
|---|---|---|

CFMIP-2

| Page 3: Inserted | mark.webb | 27/10/2016 10:08:00 |
|---|---|---|

CFMIP-2/CMIP5

| Page 3: Deleted | mark.webb | 27/10/2016 10:43:00 |
|---|---|---|

 in CMIP5

| Page 4: Deleted | mark.webb | 27/10/2016 10:05:00 |
|---|---|---|

CMIP5/CFMIP-2

| Page 4: Inserted | mark.webb | 27/10/2016 10:08:00 |
|---|---|---|

CFMIP-2/CMIP5

| Page 4: Inserted | mark.webb | 27/10/2016 10:44:00 |
|---|---|---|

CFMIP-3/

| Page 4: Inserted | mark.webb | 27/10/2016 10:45:00 |
|---|---|---|

-3

| Page 4: Inserted | mark.webb | 27/10/2016 12:02:00 |
|---|---|---|

 (Given the names of other CMIP6 experiments this experiment might have been better named *amip-4xCO2-rad,* but this inconsistency was only noticed after the experiment names were finalised and propagated to the ESGF).

| Page 4: Formatted | mark.webb | 27/10/2016 12:06:00 |
|---|---|---|

Font: Not Italic

| Page 4: Formatted | mark.webb | 27/10/2016 12:06:00 |
|---|---|---|

Font: Not Italic

| Page 4: Formatted | mark.webb | 27/10/2016 12:06:00 |
|---|---|---|

Font: Not Italic

| Page 4: Formatted | mark.webb | 27/10/2016 12:06:00 |
|---|---|---|

Font: Not Italic

| Page 5: Inserted | mark.webb | 27/10/2016 10:46:00 |
|---|---|---|

informally

| Page 5: Deleted | mark.webb | 27/10/2016 10:46:00 |
|---|---|---|

CFMIP

| Page 7: Inserted | mark.webb | 27/10/2016 10:47:00 |
|---|---|---|

informally by CFMIP-3

| Page 7: Deleted | mark.webb | 27/10/2016 10:47:00 |
|---|---|---|

via CFMIP

| Page 7: Inserted | mark.webb | 27/10/2016 11:15:00 |
| --- | --- | --- |

We also consider the time variation of feedbacks in *abrupt-4xCO2* experiments to be an important area to be investigated, as this can have a substantial impact on estimates of equilibrium sensitivity (e.g. Geoffroy et al., 2013). Andrews et al., 2015 investigated such effects using two atmosphere-only GCMs forced with SSTs and sea ice from their own *abrupt-4xCO2* experiments, and attributed the time variation in the feedbacks to changes in the pattern of surface warming. Pilot studies are ongoing to develop similar experiments based on a composite SST pattern response more representative of the CMIP5 ensemble mean. We plan to organise an informal pilot intercomparison based on this within CFMIP-3 and may subsequently propose these experiments as an extension to the CFMIP-3/CMIP6 experiment set.

| Page 7: Formatted | mark.webb | 27/10/2016 11:18:00 |
| --- | --- | --- |

Font: Italic

| Page 7: Formatted | mark.webb | 27/10/2016 11:18:00 |
| --- | --- | --- |

Font: Italic

| Page 8: Deleted | mark.webb | 27/10/2016 10:30:00 |
| --- | --- | --- |

CFMIP-3

| Page 8: Inserted | mark.webb | 27/10/2016 10:30:00 |
| --- | --- | --- |

CFMIP-3/CMIP6

| Page 8: Inserted | mark.webb | 27/10/2016 10:05:00 |
| --- | --- | --- |

/CMIP6

| Page 8: Inserted | mark.webb | 27/10/2016 10:49:00 |
| --- | --- | --- |

/CMIP5

| Page 8: Deleted | mark.webb | 27/10/2016 10:06:00 |
| --- | --- | --- |

CFMIP-2

| Page 8: Inserted | mark.webb | 27/10/2016 10:08:00 |
| --- | --- | --- |

CFMIP-2/CMIP5

| Page 9: Deleted | mark.webb | 27/10/2016 10:30:00 |
| --- | --- | --- |

CFMIP-3

| Page 9: Inserted | mark.webb | 27/10/2016 10:30:00 |
| --- | --- | --- |

CFMIP-3/CMIP6

| Page 9: Deleted | mark.webb | 27/10/2016 10:31:00 |
| --- | --- | --- |

CFMIP-3

| Page 9: Inserted | mark.webb | 27/10/2016 10:31:00 |
| --- | --- | --- |

CFMIP-3/CMIP6

| Page 9: Deleted | mark.webb | 27/10/2016 10:08:00 |
| --- | --- | --- |

CFMIP-2

| Page 9: Inserted | mark.webb | 27/10/2016 10:08:00 |
| --- | --- | --- |

CFMIP-2/CMIP5

| Page 9: Deleted | mark.webb | 27/10/2016 10:31:00 |
| --- | --- | --- |

the CFMIP-3

| Page 9: Inserted | mark.webb | 27/10/2016 10:31:00 |
| --- | --- | --- |

CFMIP-3/CMIP6

| Page 9: Deleted | mark.webb | 27/10/2016 10:31:00 |
| --- | --- | --- |

CFMIP-3

| | | |
|---|---|---|
| **Page 9: Inserted** | **mark.webb** | **27/10/2016 10:31:00** |

CFMIP-3/CMIP6

| | | |
|---|---|---|
| **Page 9: Deleted** | **mark.webb** | **27/10/2016 10:31:00** |

CFMIP-3

| | | |
|---|---|---|
| **Page 9: Inserted** | **mark.webb** | **27/10/2016 10:31:00** |

CFMIP-3/CMIP6

| | | |
|---|---|---|
| **Page 9: Deleted** | **mark.webb** | **27/10/2016 10:31:00** |

CFMIP-3

| | | |
|---|---|---|
| **Page 9: Inserted** | **mark.webb** | **27/10/2016 10:31:00** |

CFMIP-3/CMIP6

| | | |
|---|---|---|
| **Page 9: Deleted** | **mark.webb** | **27/10/2016 10:31:00** |

CFMIP-3

| | | |
|---|---|---|
| **Page 9: Inserted** | **mark.webb** | **27/10/2016 10:31:00** |

CFMIP-3/CMIP6

| | | |
|---|---|---|
| **Page 9: Deleted** | **mark.webb** | **27/10/2016 10:31:00** |

CFMIP-3

| | | |
|---|---|---|
| **Page 9: Inserted** | **mark.webb** | **27/10/2016 10:31:00** |

CFMIP-3/CMIP6

| | | |
|---|---|---|
| **Page 9: Deleted** | **mark.webb** | **27/10/2016 10:31:00** |

CFMIP-3

| | | |
|---|---|---|
| **Page 9: Inserted** | **mark.webb** | **27/10/2016 10:31:00** |

CFMIP-3/CMIP6

| | | |
|---|---|---|
| **Page 9: Deleted** | **mark.webb** | **27/10/2016 10:06:00** |

CFMIP-2

| | | |
|---|---|---|
| **Page 9: Inserted** | **mark.webb** | **27/10/2016 10:08:00** |

CFMIP-2/CMIP5

| | | |
|---|---|---|
| **Page 9: Deleted** | **mark.webb** | **27/10/2016 10:31:00** |

CFMIP-3

| | | |
|---|---|---|
| **Page 9: Inserted** | **mark.webb** | **27/10/2016 10:31:00** |

CFMIP-3/CMIP6

| | | |
|---|---|---|
| **Page 9: Deleted** | **mark.webb** | **27/10/2016 10:32:00** |

CFMIP-3

| | | |
|---|---|---|
| **Page 9: Inserted** | **mark.webb** | **27/10/2016 10:32:00** |

CFMIP-3/CMIP6

| | | |
|---|---|---|
| **Page 10: Deleted** | **mark.webb** | **27/10/2016 10:32:00** |

CFMIP-3

| | | |
|---|---|---|
| **Page 10: Inserted** | **mark.webb** | **27/10/2016 10:32:00** |

CFMIP-3/CMIP6

| | | |
|---|---|---|
| **Page 10: Deleted** | **mark.webb** | **27/10/2016 10:32:00** |

CFMIP-3

| Page 10: Inserted | mark.webb | 27/10/2016 10:32:00 |
|---|---|---|

CFMIP-3/CMIP6

| Page 10: Deleted | mark.webb | 27/10/2016 10:32:00 |
|---|---|---|

CFMIP-3

| Page 10: Inserted | mark.webb | 27/10/2016 10:32:00 |
|---|---|---|

CFMIP-3/CMIP6

| Page 10: Deleted | mark.webb | 27/10/2016 10:32:00 |
|---|---|---|

CFMIP-3

| Page 10: Inserted | mark.webb | 27/10/2016 10:32:00 |
|---|---|---|

CFMIP-3/CMIP6,

| Page 11: Deleted | mark.webb | 27/10/2016 10:33:00 |
|---|---|---|

CFMIP-3

| Page 11: Inserted | mark.webb | 27/10/2016 10:33:00 |
|---|---|---|

CFMIP-3/CMIP6

| Page 11: Deleted | mark.webb | 27/10/2016 10:07:00 |
|---|---|---|

CFMIP-2

| Page 11: Inserted | mark.webb | 27/10/2016 10:08:00 |
|---|---|---|

CFMIP-2/CMIP5

| Page 11: Inserted | mark.webb | 27/10/2016 10:54:00 |
|---|---|---|

-3/CMIP6

| Page 11: Inserted | mark.webb | 27/10/2016 10:55:00 |
|---|---|---|

-3/CMIP6

| Page 11: Inserted | mark.webb | 27/10/2016 10:55:00 |
|---|---|---|

-

| Page 11: Deleted | mark.webb | 27/10/2016 10:55:00 |
|---|---|---|

e

| Page 11: Inserted | mark.webb | 27/10/2016 10:56:00 |
|---|---|---|

e

| Page 11: Inserted | mark.webb | 27/10/2016 10:56:00 |
|---|---|---|

-3/CMIP6

| Page 12: Inserted | mark.webb | 27/10/2016 10:56:00 |
|---|---|---|

CFMIP-3/CMIP6

| Page 12: Inserted | mark.webb | 27/10/2016 10:58:00 |
|---|---|---|

CFMIP-2/

| Page 12: Inserted | mark.webb | 27/10/2016 10:58:00 |
|---|---|---|

CFMIP-3/CMIP6

| Page 12: Deleted | mark.webb | 27/10/2016 10:08:00 |
|---|---|---|

CFMIP-2

| Page 12: Inserted | mark.webb | 27/10/2016 10:08:00 |
|---|---|---|

CFMIP-2

| Page 12: Inserted | mark.webb | 27/10/2016 10:59:00 |
|---|---|---|

-3/CMIP6

| | | |
|---|---|---|
| **Page 12: Deleted** | **mark.webb** | **27/10/2016 10:59:00** |

 in CMIP6

| | | |
|---|---|---|
| **Page 12: Inserted** | **mark.webb** | **27/10/2016 10:59:00** |

CFMIP-2/

| | | |
|---|---|---|
| **Page 12: Inserted** | **mark.webb** | **27/10/2016 11:00:00** |

-2/CMIP5

| | | |
|---|---|---|
| **Page 12: Inserted** | **mark.webb** | **27/10/2016 11:00:00** |

CFMIP-3/

| | | |
|---|---|---|
| **Page 12: Inserted** | **mark.webb** | **27/10/2016 11:00:00** |

CFMIP-3/CMIP6

| | | |
|---|---|---|
| **Page 12: Inserted** | **mark.webb** | **27/10/2016 11:00:00** |

-3

| | | |
|---|---|---|
| **Page 12: Deleted** | **mark.webb** | **27/10/2016 10:33:00** |

CFMIP-3

| | | |
|---|---|---|
| **Page 12: Inserted** | **mark.webb** | **27/10/2016 10:33:00** |

CFMIP-3/CMIP6

| | | |
|---|---|---|
| **Page 13: Deleted** | **mark.webb** | **27/10/2016 10:33:00** |

CFMIP-3

| | | |
|---|---|---|
| **Page 13: Inserted** | **mark.webb** | **27/10/2016 10:33:00** |

CFMIP-3/CMIP6

| | | |
|---|---|---|
| **Page 13: Deleted** | **mark.webb** | **27/10/2016 10:33:00** |

CFMIP-3

| | | |
|---|---|---|
| **Page 13: Inserted** | **mark.webb** | **27/10/2016 10:33:00** |

CFMIP-3/CMIP6

| | | |
|---|---|---|
| **Page 13: Inserted** | **mark.webb** | **27/10/2016 11:02:00** |

informally organised

| | | |
|---|---|---|
| **Page 13: Inserted** | **mark.webb** | **27/10/2016 11:02:00** |

-3

| | | |
|---|---|---|
| **Page 13: Inserted** | **mark.webb** | **27/10/2016 11:03:00** |

CFMIP-3/

| | | |
|---|---|---|
| **Page 13: Deleted** | **mark.webb** | **27/10/2016 11:03:00** |

/CFMIP

| | | |
|---|---|---|
| **Page 13: Inserted** | **mark.webb** | **27/10/2016 11:03:00** |

-3/

| | | |
|---|---|---|
| **Page 13: Deleted** | **mark.webb** | **27/10/2016 11:03:00** |

| | | |
|---|---|---|
| **Page 13: Inserted** | **mark.webb** | **27/10/2016 11:04:00** |

CFMIP-3/

| | | |
|---|---|---|
| **Page 13: Inserted** | **mark.webb** | **27/10/2016 11:05:00** |

-3/CMIP6

| | | |
|---|---|---|
| **Page 13: Deleted** | **mark.webb** | **27/10/2016 11:05:00** |

CFMIP

| Page 14: Inserted | mark.webb | 27/10/2016 11:06:00 |
| --- | --- | --- |

the

| Page 14: Inserted | mark.webb | 27/10/2016 11:05:00 |
| --- | --- | --- |

-3/CMIP6

| Page 14: Inserted | mark.webb | 27/10/2016 11:05:00 |
| --- | --- | --- |

 DECK

| Page 14: Deleted | mark.webb | 27/10/2016 11:05:00 |
| --- | --- | --- |

| Page 14: Deleted | mark.webb | 27/10/2016 10:07:00 |
| --- | --- | --- |

CMIP5/CFMIP-2

| Page 14: Inserted | mark.webb | 27/10/2016 10:08:00 |
| --- | --- | --- |

CFMIP-2/CMIP5

| Page 14: Inserted | mark.webb | 28/10/2016 15:55:00 |
| --- | --- | --- |

ed

| Page 14: Deleted | mark.webb | 27/10/2016 10:07:00 |
| --- | --- | --- |

CFMIP-2

| Page 14: Inserted | mark.webb | 27/10/2016 10:08:00 |
| --- | --- | --- |

CFMIP-2/CMIP5

| Page 14: Inserted | mark.webb | 27/10/2016 10:08:00 |
| --- | --- | --- |

CFMIP-2/CMIP5

| Page 14: Deleted | mark.webb | 27/10/2016 10:07:00 |
| --- | --- | --- |

CMIP5/CFMIP-2

| Page 15: Inserted | mark.webb | 27/10/2016 11:07:00 |
| --- | --- | --- |

-

| Page 15: Inserted | mark.webb | 27/10/2016 11:07:00 |
| --- | --- | --- |

**-3/CMIP6**

| Page 16: Deleted | mark.webb | 27/10/2016 10:35:00 |
| --- | --- | --- |

CFMIP-3

| Page 16: Inserted | mark.webb | 27/10/2016 10:35:00 |
| --- | --- | --- |

CFMIP-3/CMIP6

| Page 17: Deleted | mark.webb | 27/10/2016 10:35:00 |
| --- | --- | --- |

CFMIP-3

| Page 17: Inserted | mark.webb | 27/10/2016 10:35:00 |
| --- | --- | --- |

CFMIP-3/CMIP6

| Page 17: Deleted | mark.webb | 27/10/2016 10:08:00 |
| --- | --- | --- |

CFMIP-2

| Page 17: Inserted | mark.webb | 27/10/2016 10:08:00 |
| --- | --- | --- |

CFMIP-2/CMIP5

| Page 18: Inserted | mark.webb | 27/10/2016 10:08:00 |
| --- | --- | --- |

-3/CMIP6

**Page 19: Inserted** | **mark.webb** | **27/10/2016 10:10:00**

-3/CMIP6

**Header and footer changes**

**Text Box changes**

**Header and footer text box changes**

**Footnote changes**

**Endnote changes**

---

## Author Response (AR3)

**Response to Topical Editor Decision: Publish as is by J. C. Hargreaves on "The Cloud Feedback Model Intercomparison Project (CFMIP) contribution to CMIP6" by Mark J. Webb et al.**

*Dear Julia,*

**In the Author's Response you wrote, "Pilot studies are ongoing to develop similar experiments based on a composite SST pattern response more representative if the CMIP5 ensemble mean." But it looks like you have correctly written "of" rather than "if" in the actual manuscript.**

*Thank you for spotting this.  We agree that the response should have said:*

*"Pilot studies are ongoing to develop similar experiments based on a composite SST pattern response more representative of the CMIP5 ensemble mean."*

*We also confirm that the text in the manuscript it correct.*

**Happy to accept the paper.**

*Thank you.*

*Regards,*
*Mark Webb*
*(On behalf of the authors)*